# FEDGP: BUFFER-BASED GRADIENT PROJECTION FOR CONTINUAL FEDERATED LEARNING

## ABSTRACT

Continual Federated Learning (CFL) is essential for enabling real-world applications where multiple decentralized clients adaptively learn from continuous data streams. A significant challenge in CFL is mitigating catastrophic forgetting, where models lose previously acquired knowledge when learning new information. Existing works on this issue either make unrealistic assumptions about the availability of task boundaries or heavily rely on surrogate samples. To address this limitation, we introduce a buffer-based Gradient Projection method (FedGP). This method tackles catastrophic forgetting by leveraging local buffer samples and aggregated buffer gradients, thus preserving knowledge across multiple clients. Our method is compatible with various existing continual learning and CFL techniques, enhancing their performance in the CFL context. Our experiments on standard benchmarks show consistent performance improvements across diverse scenarios. For example, on a task-incremental learning setting with CIFAR100, our method can help increase the accuracy up to 27%. Our code is available at `https://anonymous.4open.science/r/FedGP-F8D4`.

## 1 INTRODUCTION

Federated Learning (FL) is a machine learning technique that facilitates collaborative model training among a large number of users while keeping data decentralized for privacy and efficient communication. In real-world applications, models trained via FL need the flexibility to continuously adapt to new data streams without forgetting past knowledge. This is critical in a variety of scenarios, such as autonomous vehicles, which must adapt to changes in the surroundings like new buildings or vehicle types without losing proficiency in previously encountered contexts. These real-world considerations make it essential to integrate FL with continual learning (CL) (Shmelkov et al., 2017; Chaudhry et al., 2018; Thrun, 1995; Aljundi et al., 2017; Chen & Liu, 2018; Aljundi et al., 2018), thereby giving rise to the concept of Continual Federated Learning (CFL).

The biggest challenge in CFL, as in CL, is *catastrophic forgetting*, where the model gradually shifts its focus from old data to new data and unintentionally discards previously acquired knowledge. Initial attempts to mitigate catastrophic forgetting in CFL incorporated existing CL solutions at each client of FL, such as replaying previous task data or penalizing the updates of weights that are crucial for preserving the knowledge from earlier tasks. However, recent works (Bakman et al., 2023; Ma et al., 2022; Yoon et al., 2021) have observed that this naïve approach cannot fully mitigate the problem due to two reasons: (i) small-scale devices participating in FL only have limited buffer size to store the data from previous tasks, (ii) data distributions are not identical across clients in FL. Moreover, existing methods developed for CFL suffer from several limitations. These include scalability issues as the number of tasks grows (Yoon et al., 2021; Venkatesha et al., 2022), the need for significant effort in generating or collecting surrogate data (Ma et al., 2022), and significant communication overhead (Yao & Sun, 2020). A crucial constraint shared by all these methods is that they require explicit task boundaries. Mitigating catastrophic forgetting in practical scenarios where fixed task boundaries are absent throughout the training process, known as *general continual learning* (Buzzega et al., 2020), remains an important open question.

To address these existing challenges of CFL, we introduce a method called buffer-based Gradient Projection, which we dub FedGP. Our approach, illustrated in Fig. 1, involves two key components:

1. **Global Buffer Gradients**: Each client $k$ computes the local buffer gradient $g_{\text{ref}}^k$ of the global model with respect to its local buffer data. All local buffer gradients are then securely averaged to obtain aggregated buffer gradient $g_{\text{ref}}$.

2. **Local Gradient Projection**: In the next round, each client $k$ updates its local model such that the direction for the model update does not conflict with aggregated buffer gradient $g_{\text{ref}}$ from the previous round, ensuring each client preserves past information from all clients.

Importantly, our FedGP method is designed to be fully compatible with (i) general continual learning settings, when task boundary is unknown, and (ii) secure aggregation techniques (Bonawitz et al., 2017). Secure aggregation ensures that while clients share gradients or model updates, the individual data remains private (Bakman et al., 2023).

**Our contributions**: We introduce a new method for CFL, called FedGP. This method utilizes information from previous tasks across clients to effectively mitigate catastrophic forgetting, without having access to task boundaries or surrogate samples. Furthermore, FedGP can seamlessly integrate with existing FL+CL and CFL techniques to enhance performance.

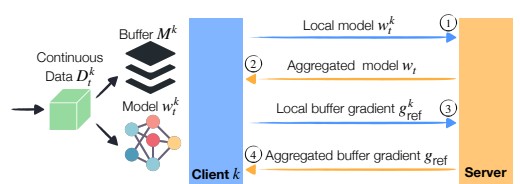

We conduct comprehensive experiments to demonstrate the effectiveness of FedGP across various standard image classification benchmarks and a text classification task on the sequential-YahooQA dataset (Zhang et al., 2015; Mehta et al., 2021). FedGP consistently improves accuracy and reduces forgetting on top of existing CL and CFL baselines across diverse benchmark datasets. Further, we evaluate the robustness of our method considering various buffer sizes, communication frequency, asynchronous environments, and different numbers of tasks and users.

Figure 1: An overview of our proposed method, FedGP. In each round $t$, client $k$ receives data $D_t^k$ and trains a local model $w_t^k$. To address catastrophic forgetting, a portion of the incoming data is stored in a buffer $\mathcal{M}^k$. Given the aggregated model $w_t$ provided by the central server, each client computes the gradient with respect to $w_t$ using its buffer data $\mathcal{M}^k$. The server securely aggregates the local buffer gradients from all clients to obtain an aggregated buffer gradient $g_{\text{ref}}$, which will guide the local model update for each client $k$ in the subsequent round.

## 2 RELATED WORK

Prior work related to our paper falls into three categories: Continual Learning (CL), Federated Learning (FL), and Continual Federated Learning (CFL). Further details are in Appendix E.

CL addresses the problem of learning multiple tasks consecutively using a single model. Catastrophic forgetting (McCloskey & Cohen, 1989; Ratcliff, 1990; French, 1999), where a classifier trained for a current task performs poorly on previous tasks, is a major challenge. Existing approaches can be categorized into regularization-based (Kirkpatrick et al., 2017; Zenke et al., 2017; Chaudhry et al., 2018; Li & Hoiem, 2017), architecture-based (Rusu et al., 2016; Yoon et al., 2017; Mallya & Lazebnik, 2018; Serra et al., 2018; Fernando et al., 2017; Wortsman et al., 2020; Pham et al., 2021; Zhao et al., 2022), and replay-based methods (Ratcliff, 1990; Robins, 1995; Rebuffi et al., 2017; Shin et al., 2017; Aljundi et al., 2019; Lopez-Paz & Ranzato, 2017; Chaudhry et al., 2019; Prabhu et al., 2020; Wu et al., 2019; Cha et al., 2021; Wang et al., 2022). Despite its simplicity, replay-based techniques have shown great performances on multiple benchmarks (Mai et al., 2022; Parisi et al., 2019). FedGP leverages a replay-based method that alleviates forgetting by reusing a portion of data from previous tasks.

FL enables collaborative training of a model with improved data privacy (Kairouz et al., 2021; Lim et al., 2020; Zhao et al., 2018; Konečnỳ et al., 2016). FedAvg (McMahan et al., 2017) is a widely used FL algorithm, but most existing methods (Li et al., 2020; Shoham et al., 2019; Karimireddy et al., 2020; Li et al., 2019; Mohri et al., 2019) assume static data distribution over time, ignoring temporal dynamics.

CFL tackles the problem of learning multiple consecutive tasks in the FL setup. FedProx (Li et al., 2020) and FedCurv (Shoham et al., 2019) aim to preserve previously learned tasks, while FedWeIT (Yoon et al., 2021) and NetTailor (Venkatesha et al., 2022) prevent interference between irrelevant tasks. Other methods including CFeD (Ma et al., 2022), FedCL (Yao & Sun, 2020), and GLFC (Dong et al., 2022) use surrogate datasets, importance weights, or class-aware techniques to distill the knowledge obtained from previous tasks. However, existing CFL methods suffer from several limitations, *e.g.*, not scalable as the number of tasks increases (Yoon et al., 2021; Venkatesha et al., 2022), requiring a surrogate dataset (Ma et al., 2022) or additional communication overhead (Yao & Sun, 2020), and not applicable to general continual setting that does not have fixed task boundaries.

## 3 PRELIMINARIES

We focus on finding a single classifier $f$ (having model parameter $w$) that performs well on $T$ tasks. We assume that at time slot $t \in [T]$, the classifier is only allowed to be trained for task $t$, where we define $[N] := \{1, \cdots, N\}$ for a positive integer $N$. We assume the feature-label samples $(x_t, y_t)$ for task $t$ are drawn from an unknown distribution $D_t$. The optimization problem for CL at time $\tau \in [T]$ is written as

$$\min_w \sum_{t=1}^{\tau} \mathbb{E}_{(x_t, y_t) \sim D_t} \left[ \ell \left( y_t, f \left( x_t; w \right) \right) \right], \tag{1}$$

where $\ell$ is the loss function, and $f(x_t; w)$ is the output of classifier $f$ with parameter $w$, for input $x_t$. We consider a practical scenario where we do not have enough storage to save all the data seen for the previous task ($t < \tau$); instead, we employ a replay buffer $\mathcal{M}$ that selectively stores a subset of data. We use the buffer data as a proxy to summarize past samples and refine the model updates. We constrain the model updates in a way that the average loss for the data in buffer $\mathcal{M}$ does not increase. Given the model $w_{\tau-1}$ trained on previous tasks, the constrained optimization problem at time $\tau \in [T]$ is represented as:

$$\min_w \mathbb{E}_{(x_\tau, y_\tau) \sim D_\tau} \left[ \ell \left( y_\tau, f \left( x_\tau; w \right) \right) \right]$$
$$\text{s.t.} \quad \mathbb{E}_{(x_b, y_b) \sim D_b} \left[ \ell \left( y_b, f \left( x_b; w \right) \right) \le \ell \left( y_b, f \left( x_b; w_{\tau-1} \right) \right) \right], \tag{2}$$

where $D_b$ is a uniform distribution over the samples in buffer $\mathcal{M}$, and $(x_b, y_b)$ are sampled from this distribution $D_b$. The optimization problem in Eq. 2 can be reformulated for various CL methods as below. First, some methods including DER (Buzzega et al., 2020) use regularization techniques to find the model parameter $w$ that minimizes the loss with respect to the local replay buffer $\mathcal{M}$ as well as current samples. For a given regularization coefficient $\gamma$, the optimization problem for CL with replay buffers at time $\tau \in [T]$ is:

$$\min_w \mathbb{E}_{(x_\tau, y_\tau) \sim D_\tau} \left[ \ell \left( y_\tau, f(x_\tau; w) \right) \right] + \gamma \mathbb{E}_{(x_b, y_b) \sim D_b} \left[ \ell \left( y_b, f(x_b; w) \right) \right]. \tag{3}$$

Second, some other methods, including A-GEM (Chaudhry et al., 2019), attempt to approximately implement the constraints of Eq. 2 by considering the gradients with respect to the current/buffer data. Specifically, the constraint promotes the alignment of the gradient with respect to the current batch of data $(x_\tau, y_\tau)$ and that for the buffer data $(x_b, y_b)$ sampled from the distribution $D_b$. This optimization problem at time $\tau \in [T]$ is formulated as:

$$\min_w \mathbb{E}_{(x_\tau, y_\tau) \sim D_\tau} \left[ \ell \left( y_\tau, f \left( x_\tau; w \right) \right) \right]$$
$$\text{s.t.} \quad \mathbb{E}_{(x_\tau, y_\tau) \sim D_\tau, (x_b, y_b) \sim D_b} \left[ \langle \nabla_w \left[ \ell \left( y_\tau, f \left( x_\tau; w \right) \right) \right], \nabla_w \left[ \ell \left( y_b, f \left( x_b; w \right) \right) \right] \rangle \right] \ge 0 \tag{4}$$

For the continual *federated* learning (CFL) setup where the data is owned by $K$ clients, we use the superscript $k \in [K]$ to denote each client, *i.e.*, client $k$ samples the data from $D_t^k$ at time $t$ and employs a local replay buffer $\mathcal{M}^k$. In the case of using FedAvg (McMahan et al., 2017), each round of the CFL is operated as follows. First, each client $k \in [K]$ performs multiple iterations of local updates with $D_t^k$ with the assistance of replay buffer $\mathcal{M}^k$. Second, once the local training is completed, each client sends the model updates to the central server. Finally, the central server aggregates the model updates and transmits them back to clients.

## 4 FEDGP

We introduce a method FedGP that is compatible with various CL and CFL techniques, significantly enhancing their performance in the CFL context. Our approach draws inspiration from A-GEM (Chaudhry et al., 2019), which projects the gradient with respect to its own historical data. Building upon this idea, we utilize the global buffer gradient, which is the average buffer gradient across all clients, as a reference to project the local gradient. This allows us to take advantage of the collective experience of multiple clients and mitigate the risk of forgetting previously learned knowledge in FL scenarios.

---

**Algorithm 1** FedAvg ServerUpdate with FedGP

---

Initialize random $w^k$, and set $\mathcal{M}^k = \{\}, g_{\text{ref}} = $ None
**for** each task $t = 1$ **to** $T$ **do**
  **for** each communication $r = 1$ **to** $R$ **do**
    $w^k \leftarrow$ ClientUpdate$(t, w^k, g_{\text{ref}})$, $\forall k$
    $w \leftarrow$ SecAgg$\left(w^k\right)$
    $g_{\text{ref}}^k \leftarrow$ ComputeBufferGrad$(w, \mathcal{M}^k)$, $\forall k$
    $g_{\text{ref}} \leftarrow$ SecAgg$\left(g_{\text{ref}}^k\right)$
  **end for**
**end for**
Return $w$, the final global model

---

**Algorithm 2** ClientUpdate$(t, w, g_{\text{ref}})$ at client $k$

---

**Input:** Task index $t$, model $w$, buffer gradient $g_{\text{ref}}$
Load the dataset $\mathcal{D}_t^k$, local buffer $\mathcal{M}^k$
Initialize $n = 0$ at the first task
**for** $(x, y) \in \mathcal{D}_t^k$ **do**
  $g = \nabla_w \left[\ell(y, f(x; w))\right]$
  $\tilde{g} \leftarrow g - \text{proj}_{g_{\text{ref}}} g \cdot \mathbf{1}(g_{\text{ref}}^\top g \leq 0)$
  $w \leftarrow w - \alpha \tilde{g}$ for some learning rate $\alpha$
  $\mathcal{M}^k \leftarrow$ ReservoirSampling$(\mathcal{M}^k, (x, y), n)$
  $n \leftarrow n + 1$
**end for**
Return $w$ to server

---

**Algorithm 3** ComputeBufferGrad$(w, \mathcal{M}^k)$

---

**Input:** global model $w$, local buffer $\mathcal{M}^k$
$(x_1, y_1) \ldots (x_m, y_m) \leftarrow$ random samples from $\mathcal{M}^k$
$g = \frac{1}{m} \sum_{i=1}^m \nabla_w \left[\ell(y_i, f(x_i; w))\right]$
Return $g$ to server

---

As a replay-based method, FedGP maintains a local buffer on each client, which is a memory buffer randomly storing a subset of data sampled from old tasks. The local buffer at client $k$ is denoted by $\mathcal{M}^k$. As the continuous data is loaded to the client, it keeps updating the buffer so that $\mathcal{M}^k$ becomes a good representative of old tasks.

Algorithm 1 provides the overview of our method in CFL setup, including the process of sharing information (model and buffer gradient) between the server and each client. For each new task $t \in [T]$, the server first aggregates the local models $w^k$ from client $k \in [K]$, getting a global model $w$. Afterwards, the server aggregates the local buffer gradient $g_{\text{ref}}^k$ (the gradient computed on the global model $w$ with respect to the local buffer $\mathcal{M}^k$) from client $k \in [K]$ to obtain a global buffer gradient $g_{\text{ref}}$. It is worth noting that the term "aggregation" in this context refers to the averaging of locally computed values across all clients. Such aggregation can be securely performed by the central server using secure aggregation (Bonawitz et al., 2017), which is denoted as "SecAgg" in Algorithm 1. Note that here we have two functions used at the client side, ClientUpdate and ComputeBufferGrad, which are given in Algorithm 2 and 3, respectively.

ClientUpdate shows how client $k$ updates its local model for task $t$. The client first loads the global model $w$ and the global buffer gradient $g_{\text{ref}}$ which are received from the server in the previous round. It also loads the local buffer $\mathcal{M}^k$ storing a subset of samples for previous tasks, and the data $\mathcal{D}_t^k$ for the current task. For new batch of data $(x, y) \in \mathcal{D}_t^k$, the client computes the gradient $g = \nabla_w \ell(y, f(x; w))$ for the model $w$. The client then compares the direction of $g$ with the direction of the global buffer gradient $g_{\text{ref}}$ received from the server. When the angle between $g$ and $g_{\text{ref}}$ is greater than 90°, it implies that while using the direction of $g$ as a reference for gradient descent may improve performance on the current task, but at the cost of degrading performance on previous tasks. To retain the knowledge on the previous tasks, we do the following: whenever $g$ and $g_{\text{ref}}$ are having a negative inner product, we project the gradient $g$ onto the global buffer gradient (which can be considered as a reference) $g_{\text{ref}}$ and remove this component from $g$, *i.e.*, define

$$\tilde{g} = g - \frac{g^T g_{\text{ref}}}{g_{\text{ref}}^T g_{\text{ref}}} g_{\text{ref}} \cdot \mathbf{1}(g_{\text{ref}}^\top g \leq 0), \tag{5}$$

following the idea suggested in (Chaudhry et al., 2019). As illustrated in Fig. 2, this projection helps prevent the model updates along the direction that is harming the performance on previous tasks.

After gradient projection, the client updates its local model $w$ by applying the gradient descent step with the updated gradient $\tilde{g}$. Finally, the client updates the contents of the buffer $\mathcal{M}^k$ by using the reservoir sampling (Vitter, 1985) written in Algorithm 4 in the Appendix. Reservoir sampling selects a random sample of $|\mathcal{M}^k|$ elements from a local input stream, while ensuring that each element has an equal probability of being included in the sample. One of the advantages of this method is that it does not require any prior knowledge of the size of the data stream.

Once the updated local models $\{w^k\}_{k=1}^K$ are transmitted to the server, the global model $w$ is securely updated on the server side, and transmitted to each client. Then, each client $k$ computes the local buffer gradient (*i.e.*, the gradient of the model $w$ with respect to the samples in the local buffer $\mathcal{M}^k$) as shown in Algorithm 3 `ComputeBufferGrad`.

After each client computes the local buffer gradient $g_{\text{ref}}^k$, the server allows the use of secure aggregation to combine these local buffer gradients and update the global buffer gradient $g_{\text{ref}}$. Secure aggregation is a well-established technique in FL that ensures the server learns nothing about individual clients' data beyond their aggregated sum. This compatibility with secure aggregation enhances the privacy safeguards of our proposed approach, effectively minimizing the risk of data leakage from individual clients. The aforementioned process takes place between each communication and is repeated $R$ times within each task. After traversing $T$ tasks, the final global model $w$ is obtained, as shown in Algorithm 1.

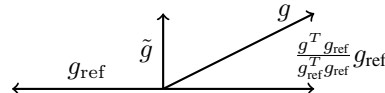

Figure 2: Illustration of the gradient projection in Eq. 5. If the angle between the gradient update $g$ and global buffer gradient (considered as a reference) $g_{\text{ref}}$ is larger than 90°, we project $g$ on $g_{\text{ref}}$ to minimize the interference and merely update along the directions of $\tilde{g}$ that is orthogonal to $g_{\text{ref}}$.

Note that the pseudocode describes the FL+FedGP process. FedGP is designed to be compatible with various CL and CFL techniques. Details of the combination are elaborated upon in the Appendix D. We have conducted extensive experiments for ablations on our algorithm design, which is decomposed into (1) gradient manipulation algorithm, and (2) buffer updating algorithm. Regarding the gradient manipulation algorithm, we tested on 8 different methods that use the reference gradient to manipulate the gradient, details of which are provided in Appendix A.1. Regarding the buffer updating algorithm, we compared three different methods (which are explained in Table 9 and Fig. 3).

## 5 EXPERIMENTS

In this section, we assess the efficacy of our method, FedGP, in combination with various CL and CFL baselines, under non-IID data distribution across clients. To evaluate these methods, we conduct experiments on image classification tasks for benchmark datasets including rotated-MNIST (Lopez-Paz & Ranzato, 2017), permuted-MNIST (Goodfellow et al., 2013), sequential-CIFAR10, and sequential-CIFAR100 (Lopez-Paz & Ranzato, 2017) datasets, as well as a text classification task (Mehta et al., 2021) on sequential-YahooQA dataset (Zhang et al., 2015). We also explore FedGP on an object detection task on a streaming CARLA dataset (Dai et al., 2023; Dosovitskiy et al., 2017) in Appendix B. All experiments were conducted on a Linux workstation equipped with 8 NVIDIA GeForce RTX 2080Ti GPUs and averaged across five runs, each using a different seed. For further details and additional results, please refer to Appendix A.

### 5.1 IMAGE CLASSIFICATION

#### 5.1.1 SETTINGS

**Evaluation Datasets.** We evaluate our approach on three CL scenarios: domain incremental learning (domain-IL), class incremental learning (class-IL), and task incremental learning (task-IL). For domain-IL, the data distribution of each class changes across different tasks. We use the rotated-MNIST and permuted-MNIST datasets for domain-IL, where each task rotates the training digits by a random angle or applies a random permutation. We create $T = 10$ tasks for domain-IL experiments.

For class-IL and task-IL, we partition the set of classes into disjoint subsets and assign each subset to a particular task. For instance, in our image classification experiments for class-IL and task-IL, we divide the CIFAR-100 dataset (with $C = 100$ classes) into $T = 10$ subsets, each of which contains the samples for $C/T = 10$ classes. Each task $t \in [T]$ is defined as the classification of images from each subset $t \in [T]$. The difference between class-IL and task-IL is that in the task-IL setup, we assume the task identity $t$ is given at inference time. That is, the model $f$ predicts among the $C/T = 10$ classes corresponding to task $t$. The class-IL and task-IL settings for CIFAR-10 are defined by splitting the CIFAR-10 dataset into $T = 5$ tasks, with each task having two unique classes.

In the FL setup, we assume that the data distribution is non-IID across the different clients. Once we define the data for each task, we split it among $K$ clients in a non-IID manner. For the rotated-MNIST or permuted-MNIST dataset, each client receives samples for two MNIST digits. To create a sequential-CIFAR10 or sequential-CIFAR100 dataset, we use the Latent Dirichlet Allocation (LDA) (Hsu et al., 2019). This algorithm partitions the dataset among multiple clients by assigning samples of each class to different clients based on the probability distribution $p \sim \text{Dir}(\alpha)$, where $\alpha = 0.3$. Communication of models and buffer gradients occurs whenever all clients complete $E$ local epochs training.

**Architecture and Hyperparameters.** For the rotated-MNIST and permuted-MNIST dataset, we use a simple CNN architecture (McMahan et al., 2017), and split the dataset into $K = 10$ clients. Each client performs local training for $E = 1$ epoch between communications, and we set the number of communication rounds as $R = 20$ for each task. For the sequential-CIFAR10 and sequential-CIFAR100 dataset, we use a ResNet18 architecture, and divide the dataset into $K = 10$ clients. Each client trains for $E = 5$ epochs between communications, and uses $R = 20$ rounds of communication for each task. During local training, Stochastic Gradient Descent (SGD) is employed with a learning rate of 0.01 for MNIST and 0.1 for CIFAR datasets. Unless otherwise noted, our studies used a 200 buffer size, a negligible storage concern for edge device like iPhone. We also conducted supplementary experiments to assess scalability by increasing the client count to $K = 20$. Additionally, we evaluated a real-world scenario where only a random subset of clients participates in training during each round. Detailed information and results are available in Appendix A.8 and A.9

**Baselines.** We evaluate the performance improvement of FedGP on three types of baselines: 1) *FL*, the basic FedAvg which trains only on the current task without considering performance on previous tasks; 2) *FL+CL*, which is FedAvg (FL) with continual learning solutions applied to clients; and 3) *CFL*, which represents the existing Continual Federated Learning methods.

CL methods include A-GEM (Chaudhry et al., 2019), which aligns model gradients for buffer and incoming data; GPM (Saha et al., 2021), using the network representation approximated by top singular vectors as the reference vector; DER (Buzzega et al., 2020), utilizing network output logits for past experience distillation; iCaRL (Rebuffi et al., 2017), which adds current task nearest-mean-of-exemplars to a memory buffer via herding and counters representation deterioration with a self-distillation loss term; and L2P (Wang et al., 2022), a state-of-the-art approach that instructs pre-trained models to sequentially learn tasks using prompt pool memory spaces.

CFL methods we tested are FedCurv (Shoham et al., 2019), which avoids updating past task-critical weights; FedProx (Li et al., 2020), introducing a proximal weight for global model alignment; CFeD (Ma et al., 2022), using surrogate dataset-based knowledge distillation; and GLFC (Dong et al., 2022), a tripartite method to counteract forgetting: 1) clients retain old task data and incorporate it with a normalization factor during new task training, 2) clients save the prior model, compute the KL divergence loss between the new and old model outputs (from the last layer), and 3) an additional proxy server is used to gather perturbed gradients for client sample generation; FOT (Bakman et al., 2023) also projects the gradients on the subspace specified by previous tasks. Details are in Appendix A.4. As for FedWeIT (Yoon et al., 2021), we believe it might not serve as a suitable benchmark given its focus on personalized FL without a global model accuracy to contrast with.

Note that CFeD, GLFC, GPM, and iCaRL require task boundaries during training. They exploit task changes to snapshot the network, with iCaRL further relying on these for memory buffer updates. For specific parameters and implementation unique to each method, please refer to Appendix A.12.

**Performance Metrics.** We assess the performance of the global model on the test dataset, which is a union of the test data for all previous tasks. The average accuracy (measured after training on task $t$) is denoted as $\text{Acc}_t = \frac{1}{t} \sum_{i=1}^{t} a_{t,i}$, where $a_{t,i}$ is evaluated on task $i$ after training

up to task $t$. Additionally, we measure a performance metric called *forgetting* in Appendix A.6, which is defined as the difference between the best accuracy obtained throughout the training and the current accuracy (Chaudhry et al., 2018). This metric measures the model's ability to retain knowledge of previous tasks while learning new ones. The forgetting at task $t$ is defined as: $\text{Fgt}_t = \frac{1}{t-1} \sum_{i=1}^{t-1} \max_{j=1,\cdots,t-1} (a_{j,i} - a_{t,i})$. We also computed the Backward transfer (BWT) and Forward transfer (FWT) metrics (Lopez-Paz & Ranzato, 2017). See Appendix A.10 for details.

### 5.1.2 RESULTS

Table 1: Average accuracy $\text{Acc}_T$ (%) on standard benchmark datasets. '-' indicates experiments we were unable to run, because of compatibility issues (e.g. GLFC and iCaRL in Domain-IL) or the absence of surrogate (e.g. CFeD on MNIST). The results, averaged over 5 random seeds, demonstrate the benefits of our proposed method in combination with all baselines. A buffer size of 200 is utilized whenever methods require it. Note that FL+L2P needs additional pretrained ViT.

| Method | rotated-MNIST (*Domain-IL*) w/o FedGP | w/ FedGP | sequential-CIFAR10 (*Class-IL*) w/o FedGP | w/ FedGP | sequential-CIFAR10 (*Task-IL*) w/o FedGP | w/ FedGP |
|---|---|---|---|---|---|---|
| FL (McMahan et al., 2017) | $68.02^{\pm3.1}$ | $79.46^{\pm4.1}$ (↑**11.44**) | $17.44^{\pm1.3}$ | $18.02^{\pm0.6}$ (↑**0.58**) | $70.58^{\pm4.0}$ | $80.83^{\pm2.0}$ (↑**10.25**) |
| FL+A-GEM (Chaudhry et al., 2019) | $68.34^{\pm5.6}$ | $74.74^{\pm2.3}$ (↑**6.40**) | $17.82^{\pm0.9}$ | $19.44^{\pm0.9}$ (↑**1.62**) | $77.14^{\pm3.1}$ | $83.16^{\pm1.6}$ (↑**6.02**) |
| FL+GPM (Saha et al., 2021) | $74.42^{\pm6.4}$ | - | $17.59^{\pm0.4}$ | - | $74.50^{\pm3.6}$ | - |
| FL+DER (Buzzega et al., 2020) | $57.73^{\pm3.6}$ | $81.33^{\pm3.3}$(↑**23.60**) | $18.44^{\pm3.7}$ | $30.94^{\pm3.8}$ (↑**12.50**) | $69.34^{\pm3.2}$ | $77.99^{\pm0.8}$ (↑**8.65**) |
| FL+iCaRL (Rebuffi et al., 2017) | - | - | $28.54^{\pm3.8}$ | $33.92^{\pm3.0}$ (↑**5.38**) | $80.85^{\pm2.9}$ | $80.09^{\pm4.1}$ (↓**0.76**) |
| FL+L2P (Wang et al., 2022) | $80.90^{\pm3.3}$ | $85.05^{\pm0.7}$ (↑**4.15**) | $28.61^{\pm1.0}$ | $81.86^{\pm7.2}$ (↑**53.25**) | $98.49^{\pm0.1}$ | $98.63^{\pm0.3}$ (↑**0.14**) |
| FedCurv (Shoham et al., 2019) | $68.21^{\pm2.6}$ | $80.53^{\pm4.3}$ (↑**12.32**) | $17.36^{\pm0.7}$ | $17.86^{\pm0.5}$ (↑**0.50**) | $67.77^{\pm1.4}$ | $81.28^{\pm1.1}$ (↑**13.51**) |
| FedProx (Li et al., 2020) | $67.79^{\pm3.2}$ | $78.74^{\pm4.1}$ (↑**10.95**) | $16.67^{\pm2.7}$ | $17.97^{\pm0.8}$ (↑**1.30**) | $69.57^{\pm6.5}$ | $81.23^{\pm1.3}$ (↑**11.66**) |
| CFeD (Ma et al., 2022) | - | - | $16.30^{\pm4.6}$ | $24.07^{\pm8.5}$ (↑**7.77**) | $77.35^{\pm4.6}$ | $79.30^{\pm5.7}$ (↑**1.95**) |
| GLFC (Dong et al., 2022) | - | - | $41.42^{\pm1.3}$ | $41.61^{\pm1.3}$ (↑**0.19**) | $81.84^{\pm2.1}$ | $82.87^{\pm1.0}$ (↑**1.03**) |

| Method | permuted-MNIST (*Domain-IL*) | | sequential-CIFAR100 (*Class-IL*) | | sequential-CIFAR100 (*Task-IL*) | |
|---|---|---|---|---|---|---|
| FL | $25.92^{\pm2.1}$ | $34.23^{\pm2.7}$ (↑**8.31**) | $8.76^{\pm0.1}$ | $17.08^{\pm1.8}$ (↑**8.32**) | $47.74^{\pm1.2}$ | $74.71^{\pm0.9}$ (↑**26.97**) |
| FL+A-GEM | $33.43^{\pm1.4}$ | $39.09^{\pm3.5}$ (↑**5.66**) | $8.90^{\pm0.1}$ | $19.53^{\pm1.3}$ (↑**10.63**) | $63.84^{\pm0.8}$ | $74.84^{\pm0.5}$ (↑**11.00**) |
| FL+GPM | $31.92^{\pm3.4}$ | - | $8.18^{\pm0.1}$ | - | $54.48^{\pm1.4}$ | - |
| FL+DER | $19.79^{\pm1.7}$ | $38.81^{\pm2.0}$ (↑**19.02**) | $13.32^{\pm1.6}$ | $22.96^{\pm3.6}$ (↑**9.64**) | $57.71^{\pm1.2}$ | $65.57^{\pm1.9}$ (↑**7.86**) |
| FL+iCaRL | - | - | $21.76^{\pm1.1}$ | $27.44^{\pm1.2}$ (↑**5.68**) | $69.91^{\pm0.7}$ | $72.83^{\pm0.5}$ (↑**2.92**) |
| FL+L2P | $66.98^{\pm4.6}$ | $69.15^{\pm3.1}$ (↑**2.17**) | $23.12^{\pm1.7}$ | $46.16^{\pm0.4}$ (↑**23.04**) | $94.46^{\pm0.4}$ | $94.91^{\pm0.2}$ (↑**0.45**) |
| FedCurv | $26.00^{\pm2.4}$ | $35.21^{\pm5.1}$ (↑**9.21**) | $8.92^{\pm0.1}$ | $16.67^{\pm0.9}$ (↑**7.76**) | $49.14^{\pm1.6}$ | $74.64^{\pm0.7}$ (↑**25.49**) |
| FedProx | $25.92^{\pm2.5}$ | $35.60^{\pm4.7}$ (↑**9.68**) | $8.75^{\pm0.2}$ | $16.92^{\pm1.4}$ (↑**8.17**) | $47.05^{\pm3.2}$ | $73.95^{\pm0.8}$ (↑**26.89**) |
| CFeD | - | - | $13.76^{\pm1.2}$ | $26.66^{\pm0.3}$ (↑**12.9**) | $51.41^{\pm1.0}$ | $72.20^{\pm0.9}$ (↑**20.79**) |
| GLFC | - | - | $13.18^{\pm0.4}$ | $13.47^{\pm0.7}$ (↑**0.29**) | $49.78^{\pm0.8}$ | $49.20^{\pm1.2}$ (↓**0.58**) |

Table 1 presents the average accuracy $\text{Acc}_T$ of various methods on image classification benchmark datasets measured upon completion of the final task $T$. For each setting, we compare the performance of an existing method with/without FedGP. We observe that the proposed methods (represented by "w/ FedGP") nearly always improves the base methods ("w/o FedGP") across the different datasets and scenarios, as seen from the upward arrows indicating performance improvements. Additional results are obtained for forgetting performance $\text{Fgt}_T$ given in Table 11 in Appendix A.6. Moreover, the performance of FedGP is analyzed progressively across tasks in Appendix A.5.

Remarkably, even a simple integration of the basic baseline, FL, with FedGP surpassed the performance of most FL+CL and CFL baselines. For instance, in the sequential-CIFAR100 experiment, FL with FedGP (17.08% class-IL, 74.71% task-IL) outperformed a majority of the baselines. Specifically, it exceeds the performance of the two advanced CFL baselines: GLFC (13.18% class-IL, 49.78% task-IL) and CFeD (13.76% class-IL, 51.41% task-IL). This underscores the substantial capability of our method in the CFL setting. Importantly, FedGP can achieve competitive performance even without utilizing information about task boundaries, unlike CFeD, GLFC, GPM, and iCaRL.

We also note that the FL+L2P method consistently exhibited the highest accuracy, largely due to the utilization of a pretrained Vision Transformer (ViT) (Dosovitskiy et al., 2020; Zhang et al., 2022), which helps mitigate the catastrophic forgetting. This is why we wrote the numbers in gray with a caveat in the caption. Yet, our approach still managed to achieve improved performance.

**Effect of Buffer size.** Table 2 reports the performances of baseline CL methods (A-GEM and DER) with/without FedGP for different buffer sizes, ranging from 200 to 5120. For all different datasets and all IL settings, increasing the buffer size further improves the advantage of applying FedGP, by providing more data for replay and mitigating forgetting. However, a finite buffer cannot maintain

Table 2: Impact of the buffer size on $\text{Acc}_T$ (%)

| Buffer Size | Method | rotated-MNIST (*Domain-IL*) | | sequential-CIFAR100 (*Class-IL*) | | sequential-CIFAR100 (*Task-IL*) | |
| --- | --- | --- | --- | --- | --- | --- | --- |
| | | w/o FedGP | w/ FedGP | w/o FedGP | w/ FedGP | w/o FedGP | w/ FedGP |
| 200 | | $68.34^{\pm5.6}$ | $74.74^{\pm2.3}$ (↑**6.40**) | $8.90^{\pm0.1}$ | $19.53^{\pm1.3}$ (↑**10.63**) | $63.84^{\pm0.8}$ | $74.84^{\pm0.5}$ (↑**11.00**) |
| 500 | FL+A-GEM | $70.18^{\pm8.7}$ | $78.74^{\pm3.2}$ (↑**8.56**) | $8.87^{\pm0.1}$ | $25.89^{\pm0.9}$ (↑**17.02**) | $64.38^{\pm1.4}$ | $79.35^{\pm0.5}$ (↑**14.97**) |
| 5120 | | $69.97^{\pm3.2}$ | $79.17^{\pm4.3}$ (↑**9.20**) | $8.85^{\pm0.1}$ | $33.30^{\pm2.5}$ (↑**24.45**) | $64.99^{\pm1.5}$ | $84.52^{\pm0.3}$ (↑**19.53**) |
| 200 | | $57.73^{\pm3.6}$ | $87.13^{\pm1.1}$ (↑**29.40**) | $13.32^{\pm1.6}$ | $22.96^{\pm3.6}$ (↑**9.64**) | $57.71^{\pm1.2}$ | $65.57^{\pm1.9}$ (↑**7.86**) |
| 500 | FL+DER | $60.00^{\pm7.2}$ | $88.83^{\pm1.6}$ (↑**28.83**) | $15.44^{\pm1.5}$ | $34.87^{\pm1.7}$ (↑**19.43**) | $60.79^{\pm1.2}$ | $73.53^{\pm1.1}$ (↑**12.74**) |
| 5120 | | $58.63^{\pm3.9}$ | $89.46^{\pm1.2}$ (↑**30.83**) | $18.89^{\pm1.0}$ | $45.76^{\pm3.8}$ (↑**26.87**) | $62.77^{\pm1.5}$ | $83.41^{\pm1.3}$ (↑**20.64**) |

Table 3: Effect of communication on accuracy (%) performance, with values in brackets indicating differences from the FL baseline.

| Method | R-MNIST *Domain-IL* | P-MNIST *Domain-IL* | S-CIFAR10 *Class-IL* | S-CIFAR10 *Task-IL* | S-CIFAR100 *Class-IL* | S-CIFAR100 *Task-IL* |
| --- | --- | --- | --- | --- | --- | --- |
| FL | $68.02^{\pm3.1}$ | $27.49^{\pm2.0}$ | $17.44^{\pm1.3}$ | $70.58^{\pm4.0}$ | $8.76^{\pm0.1}$ | $47.74^{\pm1.2}$ |
| FL w/ FedGP (2× comm overhead) | $\mathbf{79.46^{\pm4.1}}$ (↑11.44) | $\mathbf{35.91^{\pm4.0}}$ (↑8.42) | $\mathbf{18.02^{\pm0.6}}$ (↑0.58) | $\mathbf{80.83^{\pm2.0}}$ (↑10.25) | $\mathbf{17.08^{\pm1.8}}$ (↑8.32) | $\mathbf{74.71^{\pm0.9}}$ (↑26.97) |
| FL w/ FedGP (equalized comm overhead) | $75.63^{\pm3.9}$ (↑7.61) | $34.96^{\pm3.2}$ (↑7.47) | $16.65^{\pm1.0}$ (↓0.79) | $78.79^{\pm2.8}$ (↑8.21) | $13.62^{\pm0.6}$ (↑4.86) | $73.96^{\pm0.4}$ (↑26.22) |
| FL w/ FedGP (0.5× comm overhead) | $76.05^{\pm4.0}$ (↑8.03) | $29.75^{\pm4.6}$ (↑2.26) | $14.30^{\pm1.3}$ (↓3.14) | $66.90^{\pm3.6}$ (↓3.68) | $13.09^{\pm0.5}$ (↑4.33) | $69.96^{\pm0.6}$ (↑22.22) |
| FL w/ FedGP (0.2× comm overhead) | $70.59^{\pm4.7}$ (↑2.57) | $15.51^{\pm2.7}$ (↓11.98) | $13.37^{\pm2.6}$ (↓4.07) | $59.75^{\pm6.4}$ (↓10.83) | $13.59^{\pm0.9}$ (↑4.83) | $59.31^{\pm1.6}$ (↑11.57) |

the entire history of data. In Fig. 4 we reported the effect of buffer size on the accuracy of old tasks. We are assuming that every client has the same buffer size. If the buffer sizes are not equal during model training, clients with bigger buffers might add more diverse data, which could make the model biased. A possible solution is to use a reweighting algorithm, which we plan to explore in the future.

**Effect of communication frequency.**    Compared with baseline methods, FedGP has extra communication overhead for transmitting the buffer gradients from each client to the server. This means that the required amount of communication is doubled for FedGP. We consider a variant of FedGP which updates the model and buffer gradient less frequently (i.e., reduce the communication rounds for each task), which has reduced communication than the vanilla FedGP. Table 3 reports the performance for different datasets, when the communication overhead is set to 2x, 1x, 0.5x and 0.2x. First, in most cases, FedGP with equalized (1x) communication overhead is outperforming FL. In addition, for most of tested datasets including R-MNIST, P-MNIST and S-CIFAR100, FedGP outperforms FL with at most 0.5x communication overhead. This means that FedGP enjoys a higher performance with less communication, in the standard benchmark datasets for continual federated learning.

**Effect of computation overhead.**    Computation overhead is also an important aspect to consider and we have conducted experiment on the actual wall-clock time measurements. Taking a CIFAR100 experiment as an example, the running time for 200 epochs for FedAvg on our device is 4068.97s. When FedGP, which is built on top of FedAvg, was used, it ran for an additional 293.26s. This indicates that it ran 7.2% longer over the same 200 epochs. The time consumed by FedGP can be divided into two parts: (i) computing the global reference gradient after each FedAvg, and (ii) projecting the gradient. In the above experiment, the reference gradient computation was done 200 times, taking a total of 49.07s, and the gradient projection was performed on 109,471 batches, which is 68.38% of the total batches, taking a total of 244.19s. Overall, FedGP is computationally efficient and affordable for edge devices.

Table 4: $\text{Acc}_T$ (%) for asynchronous task boundaries on the sequential-CIFAR100 dataset.

| Method | *Class-IL* | *Task-IL* |
| --- | --- | --- |
| FL | $16.22^{\pm1.2}$ | $59.04^{\pm1.7}$ |
| FL+A-GEM | $16.92^{\pm1.0}$ | $69.41^{\pm1.3}$ |
| FL+A-GEM+FedGP | $30.74^{\pm1.5}$ | $\mathbf{77.70^{\pm0.4}}$ |
| FL+DER | $31.95^{\pm2.6}$ | $68.28^{\pm1.5}$ |
| FL+DER+FedGP | $\mathbf{36.29^{\pm1.0}}$ | $72.02^{\pm0.7}$ |

**Asynchronous task boundaries.**    In our previous experiments, we assumed synchronous task boundaries where clients finish tasks at the same time. However, in many real-world scenarios, different clients finish each task asynchronously. Motivated by this practical setting, we conducted experiments in an asynchronous task boundary setting on sequential-CIFAR100. For every $R = 20$ communications, instead of traversing all data allocated for the current task, each client traverses exactly 500 samples allocated to it, irrespective of whether these samples come from the same task. Consequently, some clients might finish a task faster than others and move on to the next task. Thus, during each global communication, clients could be working on different tasks. This setup more closely aligns with our general continual

learning settings, when the task boundary is unknown. Table 4 shows the accuracy of each method averaged over $T$ tasks after finishing all training, under the asynchronous setting. Similar to the synchronous case, FedGP improves the accuracy of baseline methods including A-GEM and DER. Notably, we have a better performance in the asynchronous setting (see Table 4) compared with the synchronous setting (see Table 1). This might be because, in the asynchronous setting, some clients receive new tasks earlier than others, which allows the model to be exposed to more diverse data for each round, thus reducing the forgetting effect.

**Effect of the number of tasks.** We have conducted experiments with different number of tasks for each dataset. For CIFAR100, we experimented with task numbers 5 and 10, while for CIFAR10 we tested with task numbers 2 and 5. Our results in Table 5 consistently demonstrate that the FedGP algorithm provides a significant improvement in performance across all these different task numbers. An interesting observation is that as the number of tasks increases, FedGP have better performance improvement to baseline. This is because a higher number of tasks increases the likelihood of data distribution shifts and therefore the problem of catastrophic forgetting becomes more prominent. As such, FedGP, designed to handle this issue, has more opportunities to improve the learning process in such scenarios. This might also partly explain why, in Table 1, FedGP shows a generally higher improvement over the baselines on the sequential-CIFAR100 dataset compared to the sequential-CIFAR10.

Table 5: Average accuracy $\text{Acc}_T$ (%) across various task numbers.

| (# of Task, # of Classes per Task) | sequential-CIFAR10 (*Class-IL*) | | sequential-CIFAR10 (*Task-IL*) | |
|---|---|---|---|---|
| | FL | FL w/ FedGP | FL | FL w/ FedGP |
| (2, 5) | $43.53^{\pm 0.8}$ | $44.05^{\pm 0.8}$ (↑**0.52**) | $75.54^{\pm 0.6}$ | $77.52^{\pm 0.8}$ (↑**1.98**) |
| (5, 2) | $17.44^{\pm 1.3}$ | $18.02^{\pm 0.6}$ (↑**0.6**) | $70.58^{\pm 4.0}$ | $80.83^{\pm 2.0}$ (↑**10.25**) |
| | sequential-CIFAR100 (*Class-IL*) | | sequential-CIFAR100 (*Task-IL*) | |
| (5, 20) | $16.49^{\pm 0.3}$ | $22.71^{\pm 0.9}$ (↑**6.22**) | $50.60^{\pm 0.9}$ | $69.41^{\pm 0.8}$ (↑**18.81**) |
| (10, 10) | $8.76^{\pm 0.1}$ | $17.08^{\pm 1.8}$ (↑**8.32**) | $47.74^{\pm 1.2}$ | $74.71^{\pm 0.9}$ (↑**26.97**) |

## 5.2 Text Classification

In addition to image classification, we also extended the evaluation of our method on text classification task (Mehta et al., 2021). For this purpose, we utilized the YahooQA (Zhang et al., 2015) dataset which comprises texts (questions and answers), and user-generated labels representing 10 different topics. Similar to the approach taken with the CIFAR10 dataset, we partitioned the YahooQA dataset into 5 tasks, where each task consisted of two distinct classes. Within each task, we used LDA to partition data across 10 clients in a non-IID manner. To conduct the experiment, we employed a pretrained DistilBERT (Sanh et al., 2019) with linear classification layer. We freeze the DistilBERT model and only fine-tune the additional linear layer. The results of this experiment can be found in Table 6. We can observe that FedGP consistently enhances the accuracy ($\text{Acc}_T$) over baselines, particularly in class-IL scenarios.

Table 6: Average classification accuracy $\text{Acc}_T$ (%) on split-YahooQA dataset.

| Method | sequential-YahooQA (*Class-IL*) | | sequential-YahooQA (*Task-IL*) | |
|---|---|---|---|---|
| | w/o FedGP | w/ FedGP | w/o FedGP | w/ FedGP |
| FL | $17.86^{\pm 0.6}$ | $30.67^{\pm 4.4}$ (↑**12.81**) | $80.87^{\pm 1.2}$ | $88.04^{\pm 1.4}$ (↑**7.17**) |
| FL+A-GEM | $20.86^{\pm 0.3}$ | $47.02^{\pm 1.9}$ (↑**26.16**) | $87.29^{\pm 1.3}$ | $90.20^{\pm 0.2}$ (↑**2.91**) |
| FL+DER | $43.64^{\pm 2.1}$ | $54.28^{\pm 1.3}$ (↑**10.64**) | $89.57^{\pm 0.2}$ | $90.48^{\pm 0.2}$ (↑**0.91**) |

## 6 Conclusion

In this paper, we present FedGP, a novel method of using buffer data for mitigating the catastrophic forgetting issues in CFL. Specifically, we use the gradient projection method to prevent model updates that harm the performance on previous tasks. Our empirical results on benchmark datasets (rotated-MNIST, permuted-MNIST, sequential-CIFAR10 and sequential-CIFAR100) and on a text classification dataset show that FedGP improves the performance of existing CL and CFL methods.

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

# A SUPPLEMENTARY RESULTS

In this section, we furnish additional experimental outcomes that serve to further bolster the findings of our primary investigation.

## A.1 ABLATIONS ON OUR ALGORITHM DESIGN

We added extensive experimental results for ablations on our algorithm design, which is decomposed into (1) gradient manipulation algorithm, and (2) buffer updating algorithm.

First, we considered different ways of manipulating the gradient $g$, given the reference gradient $g_{\text{ref}}$. In the below table, we compared four different methods of updating $g$:

- Average: define $g \leftarrow (g + g_{\text{ref}})/2$.
- Rotate: rotate $g$ towards $g_{\text{ref}}$ while keeping the magnitude.
- Project: project $g$ to the space that is orthogonal to $g_{\text{ref}}$.
- Project & Scale: apply Project and then scale up the vector such that the magnitude is identical to the original $g$

Recall that our FedGP applies Project method only when the angle between $g$ and $g_{\text{ref}}$ is larger than 90 degree, i.e., when the reference gradient $g_{\text{ref}}$ (measured for the previous tasks) and the gradient $g$ (measured for the current task) conflicts to each other. Our intuition for such choice is, it is better to manipulate $g$ if the direction favorable for current task is conflicting with the direction favorable for previous tasks. To support that this choice is meaningful, we compared two ways of deciding when we manipulate the gradients, denoted below:

- ($> 90$): update the gradient $g$ only when the angle($g, g_{\text{ref}}$)$> 90$
- (Always): update the gradient $g$ always

We compared the performances of above choices in Table 7, for S-CIFAR100 dataset. One can confirm that our FedGP (denoted by Project ($> 90$) in the table) far outperforms all other combinations, showing that our design (doing projection for conflicting case only) is the right choice. If we check each component (Project and ($> 90$)) independently, one can check that choosing Project outperforms Average, Rotate and Project & Scale in most cases, and choosing ($> 90$) outperforms Always in all cases.

Table 7: Effect of different gradient manipulation method on the accuracy (%) of FedGP, tested on S-CIFAR100

| Method | Class-IL | Task-IL |
|---|---|---|
| FL | 8.76±0.1 | 47.74±1.2 |
| Average (Always) | 7.26±1.95 | 35.96±3.23 |
| Average ($> 90$) | 7.79±0.65 | 36.57±1.55 |
| Rotate (Always) | 7.59±0.89 | 36.15±2.83 |
| Rotate ($> 90$) | 8.41±0.78 | 38.97±1.83 |
| Project & Scale (Always) | 8.77±0.09 | 32.96±1.10 |
| Project & Scale ($> 90$) | 12.30±0.65 | 73.61±0.75 |
| Project (Always) | 8.90±0.08 | 34.00±1.98 |
| Project ($> 90$), **ours** | **17.08±1.8** | **74.71±0.9** |

We also tested whether doing the projection is helpful in all cases when angle($g, g_{\text{ref}}$) $> 90$. We considered applying the projection for $p\%$ of the cases having angle($g, g_{\text{ref}}$) $> 90$, for $p = 10, 50, 80$ and 100. Note that $p = 100\%$ case reduces to our FedGP.

Table 8 shows the effect of projection rate $p\%$ on the accuracy, tested on S-CIFAR100 dataset. In both class-IL and task-IL settings, increasing $p$ always improves the accuracy of the FedGP method. This supports that our way of projection is suitable for the continual federated learning setup.

In Table R3-2c, we compared three different buffer updating algorithms:

Table 8: Effect of projection rate $p\%$ on the accuracy (%) of FedGP, tested on S-CIFAR100

| Method | Class-IL | Task-IL |
|---|---|---|
| FL, $p = 0\%$ | 8.76±0.1 | 47.74±1.2 |
| FedGP, $p = 10\%$ | 8.82±0.07 | 54.90±1.61 |
| FedGP, $p = 50\%$ | 8.91±0.07 | 67.89±0.67 |
| FedGP, $p = 80\%$ | 10.36±0.42 | 72.73±0.74 |
| FedGP, $p = 100\%$ (Ours) | **17.08±1.8** | **74.71±0.9** |

- Random Sampling: randomly replaces a data point in the buffer with incoming new data
- Sliding Window Sampling: replaces the earliest data point in the buffer when new data arrives
- Reservoir Sampling (Ours): given $N$ (the number of observed samples up to now) and $B$ (the buffer size), we do the following
    - when $N \leq B$, we put the current sample in the buffer
    - when $N > B$, with probability $B/N < 1$, we replace a sample in the buffer with the current sample

Note that when the number of incoming data is $N$, those $N$ samples have the same probability of getting into the buffer, for the Reservoir Sampling method used in our paper. Thus, when Reservoir Sampling is used, the buffer contains approximately equal number of samples for each task (when each task has the same number of samples), throughout the continual learning process. This is the underlying intuition why we choose such buffer updating algorithm. To support this claim, we report the performance of different sampling methods in Table 9. Here, one can confirm that our sampling method is outperforming other sampling methods.

Table 9: Effect of different buffer updating algorithms on the accuracy (%) of FedGP, tested on CIFAR100

| Method | Class-IL | Task-IL |
|---|---|---|
| FL | 8.76±0.1 | 47.74±1.2 |
| Sliding Window Sampling | 8.82±0.15 | 46.16±2.38 |
| Random Sampling | 9.72±0.10 | 54.82±1.58 |
| Reservoir Sampling (**Ours**) | **17.08±1.8** | **74.71±0.9** |

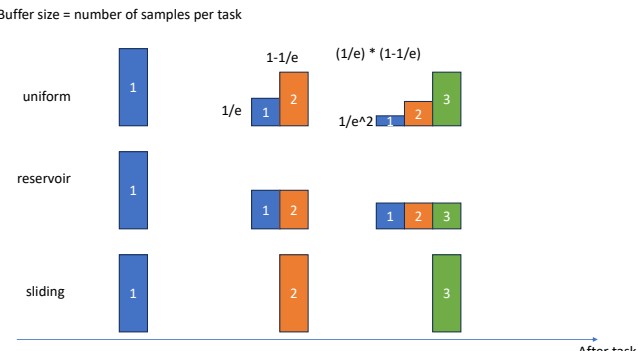

Figure 3: The sample distribution across different tasks for different buffer updating algorithms (uniform-random, sliding window, ours).

Fig. 3 compares the sample distribution across different tasks, for different buffer updating algorithms (uniform-random sampling, reservoir sampling and sliding window sampling). This clearly shows

the difference of buffer updating algorithms, which guided to the different performances reported in Table R3-2c.

All in all, the reservoir sampling method used in FedGP allows us to have balanced sample distribution across different tasks, thus allowing us to mitigate the catastrophic forgetting and to improve the accuracy in the continual federated learning setting.

## A.2 EFFECT OF BUFFER SIZE FOR OLD TASKS

Given a limited buffer size, the number of samples the buffer can maintain for old tasks is bounded above. In Fig. 4, we reported the effect of buffer size on the accuracy of the trained model for old tasks. At the end of each task, we measured the accuracy of the trained model with respect to the test data for task 1. We tested on S-CIFAR100 dataset, and considered task incremental learning (task-IL) setup.

One can observe that when the buffer size $B$ is small, the accuracy drops as the model is trained on new tasks. On the other hand, when $B \geq 100$, the task-IL accuracy for task 1 is maintained throughout the process. Note that training with our default setting $B = 200$ does not hurt the accuracy for task 1 throughout the continual learning process.

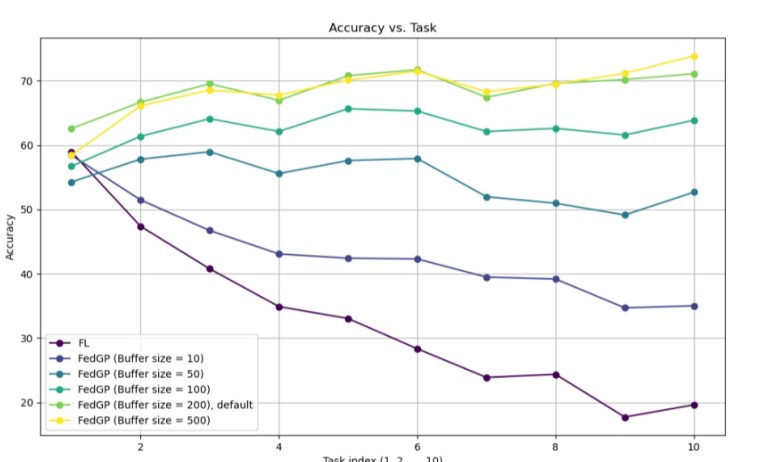

Figure 4: Accuracy (%) for Task 1, under the Task-IL setting on S-CIFAR100. We tested on FedGP with different buffer sizes.

## A.3 PERFORMANCE ON THE CURRENT TASK

Balancing the retention of old tasks and the learning of new ones is a common challenge in continual learning. It can be difficult to determine the best approach, especially when two tasks are significantly different. This is a challenge faced by many methods in continual learning.

We provided additional experimental results on the performance measured for the current task. The below Fig. 5 shows the Class-IL accuracy of FedGP (with buffer size 200) and FL for S-CIFAR100, where the total number of tasks is set to 10. During the continual learning process, we measured the accuracy of each model for the current task. One can confirm that using FedGP does not hurt the current task accuracy, compared with FL. Note that this shows that FedGP does not impair the performance of the current task, while also alleviating the forgetting in upcoming rounds.

## A.4 COMPARISON WITH FOT

We compared our method with SOTA paper (Bakman et al., 2023) proposing Federated Orthogonal Training (FOT) algorithm for continual federated learning.

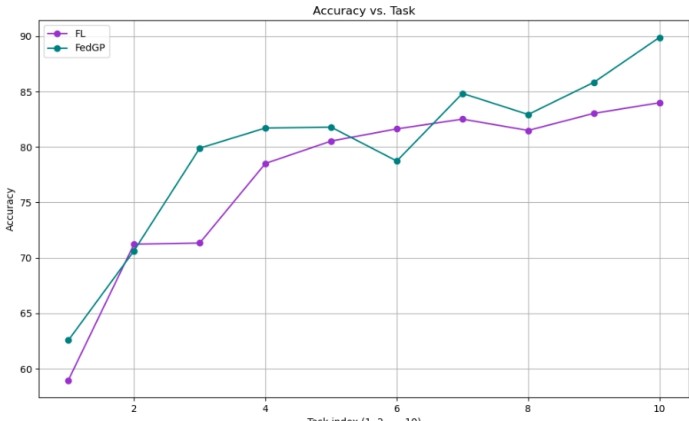

Figure 5: Class-IL Accuracy (%) of current task for FL and FedGP on the S-CIFAR100

In FOT, at the end of each task, the server aggregates the activations of each local model (computed for local data points) and computes the subspace spanned by the aggregated activations. This subspace is used during the global model update process; the gradient is updated in the direction that is orthogonal to the subspace. Note that FOT has several advantages compared with existing baselines; the privacy leakage is mitigated, the communication cost is reduced, and the solution has theoretical guarantees.

While both FOT and our FedGP project the gradients on the subspace specified by previous tasks, they have two main differences. First, the subspace is defined in a different manner: FOT relies on the representations of local model activations to define $S$. FedGP, on the other hand, relies on the gradient of model computed on its local buffer data. Second, FOT projects the gradient computed at the server side, while FedGP projects the gradient computed at each client.

The table 10 compares the accuracy of FL, FOT and FedGP (with buffer size 200) for P-MNIST and S-CIFAR100 under the task incremental learning scenario, consistent with the paper's benchmarks. Assuming that the local buffer is available, FedGP outperforms FOT.

Table 10: Comparative accuracy (%) performance analysis of FL, FOT, and FL+FedGP

| Methods | P-MNIST (Domain-IL) | S-CIFAR100 (Task-IL) |
|---|---|---|
| FL | 25.92±2.1 | 47.74±1.2 |
| FOT | 23.77±1.1 | 50.57±1.5 |
| FL+FedGP | **34.23±2.7** | **74.71±0.9** |

### A.5 PROGRESSIVE PERFORMANCE OF FedGP ACROSS TASKS

Fig. 6 depicts the average accuracy $\text{Acc}_t$ measured at task $t = 1, 2, \cdots, 10$ and the average forgetting $\text{Fgt}_t$ measured at task $t = 2, 3, \cdots, 10$. The accuracy of FedAvg rapidly drops as different tasks are given to the model, as expected. FedCurv and FedProx perform similarly to FedAvg, while A-GEM and DER partially alleviate forgetting, resulting in higher accuracies and reduced forgetting compared to FedAvg. Combining these baselines with FedGP lead to significant performance improvements, which allows the solid lines in the accuracy plot consistently remain at the top. For example, for the experiment on task-IL for sequential-CIFAR100, the accuracy measured at task 5 (denoted by $\text{Acc}_5$) is 55.37% for FedProx, while 71.12% for FedProx+FedGP. These results demonstrate that FedGP effectively mitigates forgetting and enhances existing methods in CFL.

### A.6 FORGETTING ANALYSIS ACROSS DATASETS

We present the complementary information to Table 1 in Table 11, illustrating the extent of $\text{Fgt}_T$ observed across multiple benchmark datasets. Our method exhibits exceptional effectiveness in

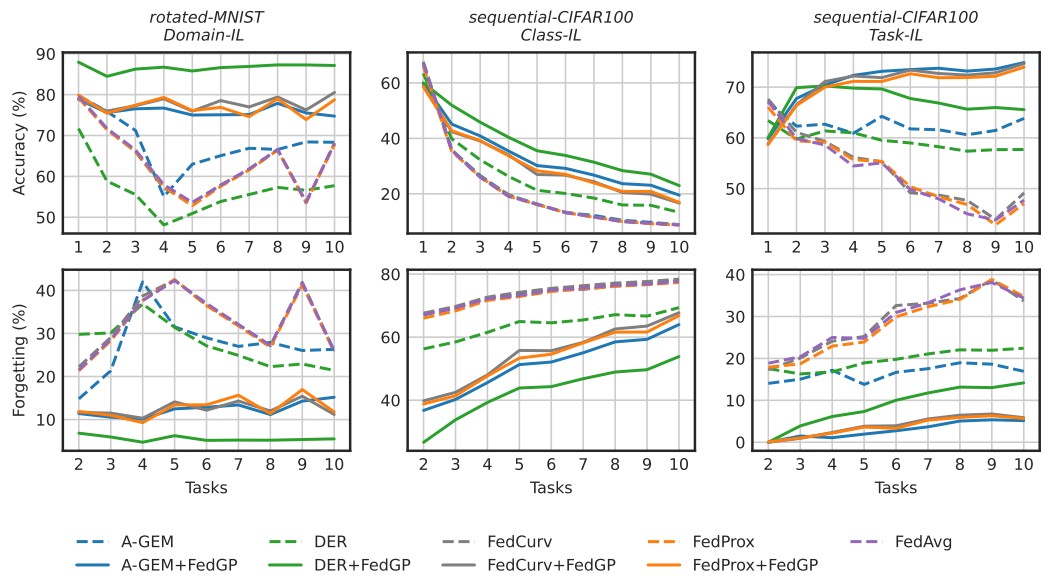

Figure 6: Evaluating accuracy ($\uparrow$) and forgetting ($\downarrow$) in multiple datasets with and without FedGP using a buffer size of 200. The solid lines indicate the results obtained with our method, while the dotted lines represent the results obtained without our method. The results show a significant improvement in accuracy as well as reduced forgetting for all settings.

mitigating forgetting. Remarkably, it demonstrates consistent performance across all datasets and baselines, making it a versatile solution.

Table 11: Average forgetting $\mathrm{Fgt}_T$ (%) (lower is better) on benchmark datasets at the final task $T$.

| | **rotated-MNIST** (*Domain-IL*) | | **sequential-CIFAR10** (*Class-IL*) | | **sequential-CIFAR10** (*Task-IL*) | |
|---|---|---|---|---|---|---|
| Method | w/o FedGP | w/ FedGP | w/o FedGP | w/ FedGP | w/o FedGP | w/ FedGP |
| FL | $25.98^{\pm3.2}$ | $11.66^{\pm2.7}(\downarrow\mathbf{14.32})$ | $80.69^{\pm3.6}$ | $78.62^{\pm4.3}(\downarrow\mathbf{2.07})$ | $15.37^{\pm4.8}$ | $4.49^{\pm1.9}(\downarrow\mathbf{10.88})$ |
| FedCurv | $25.80^{\pm2.4}$ | $11.18^{\pm2.7}(\downarrow\mathbf{14.62})$ | $80.90^{\pm6.6}$ | $79.85^{\pm3.9}(\downarrow\mathbf{1.05})$ | $19.37^{\pm4.8}$ | $4.77^{\pm1.6}(\downarrow\mathbf{14.60})$ |
| FedProx | $25.74^{\pm3.1}$ | $11.76^{\pm2.9}(\downarrow\mathbf{13.98})$ | $84.35^{\pm2.4}$ | $80.24^{\pm2.5}(\downarrow\mathbf{4.11})$ | $18.24^{\pm4.9}$ | $4.17^{\pm1.0}(\downarrow\mathbf{14.07})$ |
| FL+A-GEM | $26.30^{\pm5.7}$ | $15.18^{\pm2.4}(\downarrow\mathbf{11.12})$ | $82.18^{\pm6.6}$ | $80.38^{\pm2.5}(\downarrow\mathbf{1.80})$ | $10.00^{\pm3.0}$ | $4.15^{\pm0.7}(\downarrow\mathbf{5.85})$ |
| FL+DER | $21.42^{\pm4.0}$ | $5.51^{\pm1.2}(\downarrow\mathbf{15.91})$ | $60.98^{\pm14.6}$ | $47.88^{\pm7.2}(\downarrow\mathbf{13.10})$ | $6.34^{\pm4.9}$ | $2.73^{\pm1.3}(\downarrow\mathbf{3.61})$ |
| | **permuted-MNIST** (*Domain-IL*) | | **sequential-CIFAR100** (*Class-IL*) | | **sequential-CIFAR100** (*Task-IL*) | |
| FL | $43.47^{\pm5.3}$ | $21.40^{\pm4.9}(\downarrow\mathbf{22.07})$ | $77.69^{\pm0.5}$ | $67.02^{\pm2.3}(\downarrow\mathbf{10.67})$ | $34.38^{\pm1.6}$ | $5.39^{\pm0.8}(\downarrow\mathbf{28.99})$ |
| FedCurv | $42.88^{\pm5.0}$ | $22.85^{\pm3.5}(\downarrow\mathbf{20.03})$ | $78.40^{\pm0.9}$ | $67.75^{\pm0.8}(\downarrow\mathbf{10.65})$ | $33.71^{\pm2.2}$ | $5.86^{\pm0.7}(\downarrow\mathbf{27.85})$ |
| FedProx | $42.59^{\pm5.6}$ | $20.77^{\pm5.6}(\downarrow\mathbf{21.82})$ | $77.35^{\pm0.4}$ | $66.81^{\pm2.2}(\downarrow\mathbf{10.54})$ | $34.79^{\pm3.6}$ | $5.69^{\pm0.9}(\downarrow\mathbf{29.10})$ |
| FL+A-GEM | $35.61^{\pm5.3}$ | $24.05^{\pm2.4}(\downarrow\mathbf{11.56})$ | $77.97^{\pm0.7}$ | $63.99^{\pm2.0}(\downarrow\mathbf{13.98})$ | $16.92^{\pm1.1}$ | $5.16^{\pm0.5}(\downarrow\mathbf{11.76})$ |
| FL+DER | $45.33^{\pm5.0}$ | $34.71^{\pm5.0}(\downarrow\mathbf{10.62})$ | $69.37^{\pm1.7}$ | $53.84^{\pm6.7}(\downarrow\mathbf{15.53})$ | $22.43^{\pm0.7}$ | $14.16^{\pm1.7}(\downarrow\mathbf{8.27})$ |

### A.7 EXTENDED ANALYSIS ON THE INFLUENCE OF BUFFER SIZE

In the main body of our study, we examine the influence of different buffer sizes on the performance metric $\mathrm{Acc}_T$, utilizing rotated-MNIST and sequential-CIFAR100 datasets. To further augment our analysis, we have included two additional datasets in Table 12, incorporating various buffer sizes. By evaluating $\mathrm{Acc}_T$ (where higher values indicate better performance), we discovered that our proposed method, referred to as FedGP, consistently enhances the average accuracy across these two datasets.

### A.8 EFFECT OF THE NUMBER OF USERS.

While our previous experiments are conducted for cases with $K = 10$ clients, Table 13 shows the results for $K = 20$ clients. This results demonstrates that FedGP consistently improves the performance of baselines, across different number of clients. In line with the presentation of forgetting

Table 12: Impact of the buffer size on $\text{Acc}_T$ (%)

| Buffer Size | Method | permuted-MNIST (*Domain-IL*) | | sequential-CIFAR10 (*Class-IL*) | | sequential-CIFAR10 (*Task-IL*) | |
|---|---|---|---|---|---|---|---|
| | | w/o FedGP | w/ FedGP | w/o FedGP | w/ FedGP | w/o FedGP | w/ FedGP |
| 200 | | $33.43^{\pm1.4}$ | $39.09^{\pm3.5}$ (↑**5.66**) | $17.82^{\pm0.9}$ | $19.44^{\pm0.9}$ (↑**1.62**) | $77.14^{\pm3.1}$ | $83.16^{\pm1.6}$ (↑**6.02**) |
| 500 | FL+A-GEM | $33.35^{\pm1.0}$ | $42.45^{\pm6.9}$ (↑**9.10**) | $18.39^{\pm0.2}$ | $20.34^{\pm0.6}$ (↑**1.95**) | $78.43^{\pm3.0}$ | $85.95^{\pm0.6}$ (↑**7.52**) |
| 5120 | | $32.72^{\pm1.4}$ | $40.07^{\pm2.5}$ (↑**7.35**) | $16.41^{\pm2.6}$ | $20.64^{\pm2.2}$ (↑**4.23**) | $73.89^{\pm3.3}$ | $86.82^{\pm1.5}$ (↑**12.93**) |
| 200 | | $19.79^{\pm1.7}$ | $43.43^{\pm0.9}$ (↑**23.64**) | $18.44^{\pm3.7}$ | $30.94^{\pm3.8}$ (↑**12.50**) | $69.34^{\pm3.2}$ | $77.99^{\pm0.8}$ (↑**8.65**) |
| 500 | FL+DER | $19.17^{\pm1.6}$ | $43.38^{\pm2.4}$ (↑**24.21**) | $20.81^{\pm3.6}$ | $29.78^{\pm4.3}$ (↑**8.97**) | $71.17^{\pm1.5}$ | $74.98^{\pm3.5}$ (↑**3.81**) |
| 5120 | | $18.57^{\pm1.4}$ | $44.68^{\pm2.4}$ (↑**26.11**) | $34.75^{\pm2.2}$ | $42.38^{\pm4.5}$ (↑**7.63**) | $78.22^{\pm2.3}$ | $81.94^{\pm1.7}$ (↑**3.72**) |

Table 13: The $\text{Acc}_T$ (%) performance measured when we have $K = 20$ users. Similar to the results for $K = 10$ in Table 1, our method improves the performance of baselines.

| Method | rotated-MNIST (*Domain-IL*) | | sequential-CIFAR10 (*Class-IL*) | | sequential-CIFAR10 (*Task-IL*) | |
|---|---|---|---|---|---|---|
| | w/o FedGP | w/ FedGP | w/o FedGP | w/ FedGP | w/o FedGP | w/ FedGP |
| FL | $62.45^{\pm8.5}$ | $76.01^{\pm4.6}$ (↑**13.56**) | $16.44^{\pm1.4}$ | $15.82^{\pm1.7}$ (↓**0.62**) | $68.18^{\pm5.3}$ | $73.45^{\pm4.3}$ (↑**5.27**) |
| FedCurv | $62.57^{\pm8.3}$ | $76.46^{\pm4.1}$ (↑**13.89**) | $17.31^{\pm0.6}$ | $14.64^{\pm3.1}$ (↓**2.67**) | $67.33^{\pm3.3}$ | $70.31^{\pm3.7}$ (↑**2.98**) |
| FedProx | $62.14^{\pm8.6}$ | $75.84^{\pm4.4}$ (↑**13.70**) | $16.37^{\pm1.1}$ | $16.15^{\pm1.3}$ (↓**0.22**) | $66.24^{\pm1.4}$ | $74.79^{\pm3.9}$ (↑**8.55**) |
| FL+A-GEM | $67.66^{\pm8.0}$ | $78.10^{\pm3.6}$ (↑**10.44**) | $16.15^{\pm1.9}$ | $17.36^{\pm0.8}$ (↑**1.21**) | $72.39^{\pm3.4}$ | $80.61^{\pm2.6}$ (↑**8.22**) |
| FL+DER | $57.33^{\pm3.2}$ | $87.84^{\pm1.5}$ (↑**30.51**) | $17.13^{\pm2.3}$ | $19.18^{\pm3.7}$ (↑**2.05**) | $70.82^{\pm1.9}$ | $77.04^{\pm2.5}$ (↑**6.22**) |
| | permuted-MNIST (*Domain-IL*) | | sequential-CIFAR100 (*Class-IL*) | | sequential-CIFAR100 (*Task-IL*) | |
| FL | $20.26^{\pm1.6}$ | $20.67^{\pm4.7}$ (↑**0.41**) | $8.61^{\pm0.1}$ | $17.47^{\pm1.1}$ (↑**8.86**) | $50.00^{\pm1.6}$ | $76.29^{\pm0.8}$ (↑**26.29**) |
| FedCurv | $20.25^{\pm1.9}$ | $23.30^{\pm5.7}$ (↑**3.05**) | $8.93^{\pm0.0}$ | $19.42^{\pm1.1}$ (↑**10.49**) | $49.83^{\pm1.4}$ | $79.58^{\pm0.6}$ (↑**29.75**) |
| FedProx | $20.19^{\pm1.4}$ | $23.78^{\pm5.2}$ (↑**3.59**) | $8.88^{\pm0.1}$ | $18.86^{\pm1.0}$ (↑**9.98**) | $50.86^{\pm1.2}$ | $78.19^{\pm0.9}$ (↑**27.33**) |
| FL+A-GEM | $24.43^{\pm2.1}$ | $23.29^{\pm3.8}$ (↓**1.14**) | $8.62^{\pm0.1}$ | $19.58^{\pm1.2}$ (↑**10.96**) | $63.02^{\pm0.6}$ | $76.23^{\pm0.6}$ (↑**13.21**) |
| FL+DER | $17.89^{\pm1.3}$ | $46.17^{\pm3.0}$ (↑**28.28**) | $11.53^{\pm0.5}$ | $26.64^{\pm2.8}$ (↑**15.11**) | $57.00^{\pm1.4}$ | $69.42^{\pm1.0}$ (↑**12.42**) |

in Table 11, we present the forgetting analysis when the number of clients is set to 20 in Table 14. Notably, our method exhibits consistent and impressive performance across varying numbers of users. It consistently proves its effectiveness regardless of the specific user count, showcasing its robustness and reliability.

Table 14: The $\text{Fgt}_T$ (%) (lower is better) performance measured when we have $K = 20$ users.

| Method | rotated-MNIST (*Domain-IL*) | | sequential-CIFAR10 (*Class-IL*) | | sequential-CIFAR10 (*Task-IL*) | |
|---|---|---|---|---|---|---|
| | w/o FedGP | w/ FedGP | w/o FedGP | w/ FedGP | w/o FedGP | w/ FedGP |
| FL | $31.00^{\pm9.5}$ | $13.45^{\pm3.6}$ (↓**17.55**) | $82.62^{\pm3.1}$ | $73.39^{\pm4.5}$ (↓**9.23**) | $17.93^{\pm2.7}$ | $6.14^{\pm4.9}$ (↓**11.79**) |
| FedCurv | $30.73^{\pm9.3}$ | $12.97^{\pm3.8}$ (↓**17.76**) | $79.55^{\pm3.8}$ | $75.38^{\pm5.3}$ (↓**4.17**) | $18.19^{\pm3.0}$ | $9.14^{\pm3.1}$ (↓**9.05**) |
| FedProx | $31.04^{\pm9.7}$ | $13.31^{\pm3.4}$ (↓**17.73**) | $82.94^{\pm1.1}$ | $78.67^{\pm4.2}$ (↓**4.27**) | $20.60^{\pm2.6}$ | $8.52^{\pm3.0}$ (↓**12.08**) |
| FL+A-GEM | $25.22^{\pm8.8}$ | $11.02^{\pm3.0}$ (↓**14.20**) | $82.39^{\pm2.4}$ | $80.25^{\pm4.1}$ (↓**2.14**) | $12.29^{\pm2.2}$ | $4.00^{\pm2.4}$ (↓**8.29**) |
| FL+DER | $28.93^{\pm6.6}$ | $5.18^{\pm1.1}$ (↓**23.75**) | $55.10^{\pm9.8}$ | $60.90^{\pm3.8}$ (↑**5.80**) | $3.20^{\pm1.6}$ | $2.71^{\pm1.7}$ (↓**0.49**) |
| | permuted-MNIST (*Domain-IL*) | | sequential-CIFAR100 (*Class-IL*) | | sequential-CIFAR100 (*Task-IL*) | |
| FL | $24.27^{\pm5.2}$ | $8.67^{\pm7.0}$ (↓**15.60**) | $73.05^{\pm0.5}$ | $62.71^{\pm0.9}$ (↓**10.34**) | $27.07^{\pm1.7}$ | $2.48^{\pm0.7}$ (↓**24.59**) |
| FedCurv | $24.02^{\pm5.4}$ | $8.10^{\pm5.4}$ (↓**15.92**) | $80.07^{\pm0.5}$ | $68.58^{\pm1.1}$ (↓**11.49**) | $34.63^{\pm1.7}$ | $3.48^{\pm0.6}$ (↓**31.15**) |
| FedProx | $23.01^{\pm5.7}$ | $5.93^{\pm5.1}$ (↓**17.08**) | $79.46^{\pm0.5}$ | $68.40^{\pm0.9}$ (↓**11.06**) | $32.82^{\pm1.4}$ | $4.13^{\pm0.7}$ (↓**28.69**) |
| FL+A-GEM | $22.12^{\pm4.9}$ | $9.45^{\pm5.4}$ (↓**12.67**) | $72.97^{\pm1.1}$ | $60.27^{\pm1.3}$ (↓**12.70**) | $12.54^{\pm1.3}$ | $2.66^{\pm0.2}$ (↓**9.88**) |
| FL+DER | $32.26^{\pm1.1}$ | $27.30^{\pm4.2}$ (↓**4.96**) | $67.07^{\pm0.8}$ | $47.74^{\pm3.8}$ (↓**19.33**) | $19.78^{\pm1.7}$ | $8.67^{\pm1.4}$ (↓**11.11**) |

## A.9 RANDOM SAMPLING

We implement a more realistic federated learning environment by applying uniform sampling techniques to randomly select the participating clients in each round. We conduct experiments on CIFAR100. A total of 50 clients is set up, and during each communication, only a random 50% of the clients participate in training. As can be seen, even in such a scenario, where our algorithm cannot update the reference gradient using the local buffer from all clients, there is still an improvement in performance using our algorithm.

Table 15: Average accuracy $\text{Acc}_T$ (%) with 50 clients and 50% client sampling rate, for sequential-CIFAR100

| Method | *Class-IL* | *Task-IL* |
|---|---|---|
| FL | $7.46 \pm 0.08$ | $43.85 \pm 1.33$ |
| FL+FedGP | $9.34 \pm 0.31 \,(\uparrow 1.88)$ | $65.76 \pm 0.48 \,(\uparrow 21.91)$ |

### A.10 BACKWARD AND FORWARD TRANSFER METRICS

Our method outperforms FedAvg (FL) in both Backward and Forward Transfer metrics across the sequential-CIFAR10 and sequential-CIFAR100 datasets, as shown in the Table 16.

Table 16: Backward and Forward Transfer ($\uparrow$) Results for sequential-CIFAR100 and sequential-CIFAR10

| Metric | Dataset | Methods | *Class-IL* | *Task-IL* |
|---|---|---|---|---|
| Backward | CIFAR100 | FL | $-78.11$ | $-36.52$ |
| Backward | CIFAR100 | FL+FedGP | $-72.24 \,(\uparrow 5.87)$ | $-3.68 \,(\uparrow 32.84)$ |
| Backward | CIFAR10 | FL | $-78.78$ | $-14.48$ |
| Backward | CIFAR10 | FL+FedGP | $-78.55 \,(\uparrow 0.23)$ | $-0.60 \,(\uparrow 13.88)$ |
| Forward | CIFAR100 | FL | $16.98$ | $16.98$ |
| Forward | CIFAR100 | FL+FedGP | $17.16 \,(\uparrow 0.18)$ | $17.48 \,(\uparrow 0.50)$ |
| Forward | CIFAR10 | FL | $12.75$ | $12.74$ |
| Forward | CIFAR10 | FL+FedGP | $12.98 \,(\uparrow 0.23)$ | $12.99 \,(\uparrow 0.25)$ |

### A.11 EFFECT OF DIFFERENT CURRICULUM.

We evaluate how the performance of FedGP changes when we shuffle the order of tasks in the continual learning. We randomly shuffle the sequential-CIFAR100 task order and label them as curriculum 1 to 4, as shown in the Table 17. Regardless of the different curriculum, FL+FedGP outperforms FedAvg.

### A.12 ADDITIONAL HYPERPARAMETERS FOR SPECIFIC METHODS

In addition to the hyperparameters discussed in the main paper, additional method-specific hyperparameters are outlined in Table 18.

## B OBJECT DETECTION

Here we test FedGP on realistic streaming data (Dai et al., 2023) which leverage two open source tools, an urban driving simulator (CARLA (Dosovitskiy et al., 2017)) and a FL framework (OpenFL (Reina et al., 2021)). As shown in Fig. 7a, CARLA provides OpenFL with a real-time collection of continuous streaming vehicle camera output data and automatic annotation about object detection. This streaming data capture the spatio-temporal dynamics of data generated from real-world applications. After loading data of vehicles from CARLA, OpenFL performs collaborative training over multiple clients.

Table 17: Average accuracy $\text{Acc}_T$ (%) across randomized curriculum in sequential-CIFAR100.

| Curriculum | Methods | *Class-IL* | *Task-IL* |
|---|---|---|---|
| 1 | FL | 8.15 | 46.25 |
| 1 | FL+FedGP | $12.10 \,(\uparrow 3.95)$ | $72.69 \,(\uparrow 26.44)$ |
| 2 | FL | 8.46 | 47.56 |
| 2 | FL+FedGP | $14.37 \,(\uparrow 5.91)$ | $73.19 \,(\uparrow 25.63)$ |
| 3 | FL | 8.82 | 45.04 |
| 3 | FL+FedGP | $12.58 \,(\uparrow 3.76)$ | $74.71 \,(\uparrow 29.67)$ |
| 4 | FL | 7.87 | 43.87 |
| 4 | FL+FedGP | $14.85 \,(\uparrow 6.98)$ | $73.74 \,(\uparrow 29.87)$ |

Table 18: Additional hyperparameters for specific methods.

| Method | Parameter | Values |
|---|---|---|
| FL+DER | Regularization Coefficient | sequential-CIFAR10 (0.3), Others (1) |
| FL+L2P | Communication Round $R$ | rotated-MNIST (5), permuted-MNIST (1), sequential-CIFAR10 (20), sequential-CIFAR100 (20) |
| CFeD | Surrogate Dataset | sequential-CIFAR10 (CIFAR100), sequential-CIFAR100 (CIFAR10) |
| | Note: No server distillation included. | |

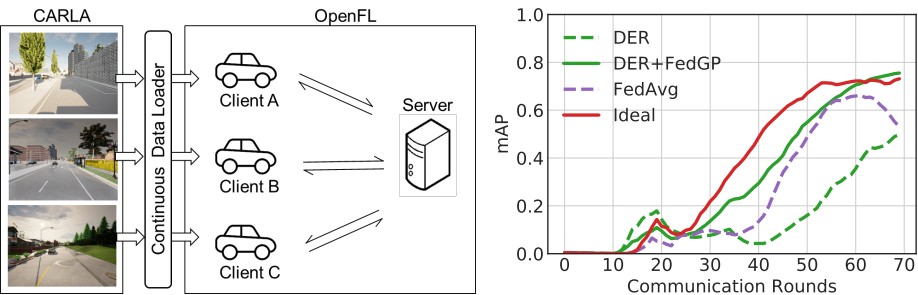

(a) Framework for automotive data evaluation.  (b) Object detection performance comparison.

Figure 7: (a) The data loader continuously supplies data from CARLA camera outputs to individual FL clients. Each client trains on its local data and updates its buffer to retain old knowledge. (b) The result shows the object detection performance comparison between Ideal, FedAvg, DER, and DER+FedGP on a realistic CARLA dataset.

We evaluate the solutions to the forgetting problem by spawning two vehicles in a virtual town. During the training of the tinyYOLO (Redmon & Farhadi, 2017) object detection model, we use a custom loss that combines classification, detection and confidence losses. In order to quantify the quality of the incremental model trained by various baselines, we report a common metric, namely, mean average precision (mAP). This metric assesses the correspondence between the detected bounding boxes and the ground truth, with higher scores indicating better performance. To calculate mAP, we analyze the prediction results obtained from pre-collected driving snippets of vehicular clients. These driving snippets are gathered by navigating the town over a duration of 3000 simulation seconds.

For those experiments on realistic CARLA streaming data, we compare the performances of Ideal, FedAvg, DER and DER+FedGP. In the Ideal scenario, the client possesses sufficient memory to retain all data from prior tasks, enabling joint training on all stored data. The last two methods are equipped with buffer size of 200. We train for 70 communication rounds and each round continues for about 200 simulation seconds. The results are presented in Fig. 7b. Note that at communication round 60, one client gets on the highway, which incurs a domain shift. One can confirm that the performance of FedAvg degrades in such domain shift scenario, whereas DER and DER+FedGP maintain the accuracy. Moreover, FedGP nearly achieves the performance of the ideal scenario with infinite buffer size, demonstrating the effectiveness of our method.

## C  ADDITIONAL ALGORITHMS

In this section, we present the pseudocode for the `ReservoirSampling` algorithm (see Algorithm 4). In the initial phase, when the buffer is not yet full (i.e., $n \leq |\mathcal{M}^k|$), `ReservoirSampling` stores each new sample $(x, y)$ in the buffer. After the buffer is full, the algorithm determines two things: (1) whether it should replace an element in the buffer with the new sample, and (2) which element in the buffer it will replace. We now prove that for reservoir sampling, the probability of a sample contained in the buffer is $\frac{|\mathcal{M}^k|}{n}$. We prove this by induction; suppose this statement holds when $n-1$ samples are observed, and we show that this holds when one more sample is observed. Note that the probability of a sample contained in the buffer can be computed as $\frac{|\mathcal{M}^k|}{n-1} * (1 - \frac{|\mathcal{M}^k|}{n} * \frac{1}{|\mathcal{M}^k|}) = \frac{|\mathcal{M}^k|}{n}$, where

- $\frac{|\mathcal{M}^k|}{n-1}$ is the probability of the sample initially contained in the buffer;
- $(1 - \frac{|\mathcal{M}^k|}{n} * \frac{1}{|\mathcal{M}^k|})$ is the probability of a sample not being kicked out of the buffer;
- $\frac{|\mathcal{M}^k|}{n} * \frac{1}{|\mathcal{M}^k|}$ is the probability of a sample being kicked out of the buffer

---

**Algorithm 4** `ReservoirSampling`$(\mathcal{M}^k, (x, y), n)$ (Vitter, 1985) at client $k$

---

**Input:** local buffer $\mathcal{M}^k$, incoming data $(x, y)$ and the number of observed samples $n$
**if** $n \leq |\mathcal{M}^k|$ **then**
    Add data $(x, y)$ into local buffer $\mathcal{M}^k$
**else**
    $i \leftarrow \text{Uniform}\{1, 2, \cdots, n\}$
    **if** $i \leq |\mathcal{M}^k|$ **then**
        $\mathcal{M}^k[i] \leftarrow (x, y)$
    **end if**
**end if**
Return $\mathcal{M}^k$, the updated local buffer

---

## D   CONTINUAL LEARNING METHODS WITH FEDGP

We provided the pseudocode for Algorithm 2 modifications when implementing FL+DER+FedGP and FL+A-GEM+FedGP, respectively presented in Algorithm 5 and Algorithm 6. Other FL+CL and CFL methods are also combined with FedGP in a similar manner.

Algorithm 5 incorporates Dark Experience Replay (DER) into the local update process on client $k \in [K]$. When the server sends the global model $w$ to client $k$, the client calculates the output logits or pre-softmax response $z$. In addition, the client samples past data $(x', y')$ and the corresponding logits $z'$ from the buffer $\mathcal{M}^k$. To address forgetting, the regularization term considers the Euclidean distance between the sampled output logits and the current model's output logits on buffer data. The gradient $g$ is then refined using this regularization term to minimize the discrepancy between the current and past output logits, thereby mitigating forgetting. The following steps are the same as in the main text.

Algorithm 6 combines with A-GEM, applying gradient projection twice. First, the client computes the gradient $g_c$ with respect to the new data from $\mathcal{D}_t^k$. After replaying previous samples $(x', y')$ stored in the local buffer $\mathcal{M}^k$, the client computes the gradient $g_b$ with respect to this buffered data. If these gradients differ significantly in terms of their direction, the client projects $g_c$ onto $g_b$ to remove interference.

## E   ADDITIONAL RELATED WORK

We summarize the prior works that are related to our paper, which are categorized as continual learning, federated learning, and continual federated learning.

### E.1   CONTINUAL LEARNING (CL)

CL is a problem of learning multiple different tasks consecutively using a single model (Ring, 1998; Delange et al., 2021). For example, when the tasks are classification problems, CL focuses on the scenario when a classifier is trained for one task in the first phase, and then trained for another task in the second phase, and so on. In general, the data loaded at the current phase has a different distribution compared to the data at the previous phases, known as domain distribution shift or class distribution shift. Unfortunately, since the learner has a limited amount of memory to store data, the classifier is only allowed to access the data for the current task, not for the previous tasks. In such a setting, catastrophic forgetting (McCloskey & Cohen, 1989; Ratcliff, 1990; French, 1999) is a notorious problem, where a classifier that performs well for the task from the current round does not perform well on the tasks from the previous rounds. There has been extensive work to address this issue and

| **Algorithm 5** DER `ClientUpdate` at client $k$ | **Algorithm 6** A-GEM `ClientUpdate` at client $k$ |
|---|---|
| **Input:** Task index $t$, model $w$, buffer gradient $g_{\text{ref}}$ 
 Load the dataset $\mathcal{D}_t^k$, local buffer $\mathcal{M}^k$ 
 Initialize $n = 0$ at the first task 
 **for** $(x, y) \in \mathcal{D}_t^k$ **do** 
 $\quad z \leftarrow h(x; w)$ where $f(x; w) \coloneqq \sigma\left(h(x; w)\right)$ 
 $\quad (x', z', y') \leftarrow \mathcal{M}^k$ 
 $\quad \ell_{\text{reg}} \leftarrow \lambda \left\| z' - h(x'; w) \right\|_2^2$ 
 $\quad g = \nabla_w \left[\ell(y, f(x; w)) + \ell_{\text{reg}}\right]$ 
 $\quad \tilde{g} \leftarrow g - \text{proj}_{g_{\text{ref}}} g \cdot \mathbf{1}(g_{\text{ref}}^\top g \leq 0)$ 
 $\quad w \leftarrow w - \alpha \tilde{g}$ for some learning rate $\alpha$ 
 $\quad$`ReservoirSampling`$(\mathcal{M}^k, (x, z, y), n)$ 
 $\quad n \leftarrow n + 1$ 
 **end for** 
 Return $w$ to server | **Input:** Task index $t$, model $w$, buffer gradient $g_{\text{ref}}$ 
 Load the dataset $\mathcal{D}_t^k$, local buffer $\mathcal{M}^k$ 
 Initialize $n = 0$ at the first task 
 **for** $(x, y) \in \mathcal{D}_t^k$ **do** 
 $\quad g_c = \nabla_w \left[\ell(y, f(x; w))\right]$ 
 $\quad (x', y') \leftarrow \mathcal{M}^k$ 
 $\quad g_b = \nabla_w \left[\ell(y', f(x'; w))\right]$ 
 $\quad g \leftarrow g_c - \text{proj}_{g_b} g_c \cdot \mathbf{1}(g_b^\top g_c \leq 0)$ 
 $\quad \tilde{g} \leftarrow g - \text{proj}_{g_{\text{ref}}} g \cdot \mathbf{1}(g_{\text{ref}}^\top g \leq 0)$ 
 $\quad w \leftarrow w - \alpha \tilde{g}$ for some learning rate $\alpha$ 
 $\quad$`ReservoirSampling`$(\mathcal{M}^k, (x, y), n)$ 
 $\quad n \leftarrow n + 1$ 
 **end for** 
 Return $w$ to server |

can be divided into three major categories: regularization-based methods, architecture-based methods and replay-based methods.

**Regularization-based methods** Some CL methods add a regularization term in the loss used for the model update; they penalize the updates on weights that are important for previous tasks. EWC (Kirkpatrick et al., 2017), SI (Zenke et al., 2017), Riemannian Walk (Chaudhry et al., 2018) are methods within this category. EWC uses Fisher information matrix to evaluate the importance of parameters for previous tasks. Besides, LwF (Li & Hoiem, 2017) leverages knowledge distillation to preserve outputs on previous tasks while learning the current task.

**Architecture-based methods** A class of CL methods assigns a subset of model parameters to each task, so that different tasks are learned by different parameters. This class of methods is also known as parameter isolation methods. Some methods including PNN (Rusu et al., 2016) and DEN (Yoon et al., 2017) uses dynamic architectures where the architecture changes dynamically as the number of tasks increases. These methods have issues where the number of required parameters grows linearly with the number of tasks. To tackle this issue, fixed network are used in the recent methods including PackNet (Mallya & Lazebnik, 2018), HAT (Serra et al., 2018) and PathNet (Fernando et al., 2017). SupSup (Wortsman et al., 2020) and DualNet (Pham et al., 2021) are the latest SOTA methods.

**Replay-based methods** To avoid catastrophic forgetting, a class of CL methods employs a replay buffer to save a small portion of the data seen in previous tasks and reuse it in the training of subsequent tasks. One of the early works in this area is ER (Ratcliff, 1990; Robins, 1995). In more contemporary studies, iCaRL (Rebuffi et al., 2017) stores exemplars of data from previous tasks and adds distillation loss for old exemplars to mitigate the forgetting issue. Deep Generative Replay (Shin et al., 2017) retains the memories of the previous tasks by loading the synthetic data generated by GANs without replaying the actual data for the previous tasks. GSS (Aljundi et al., 2019) optimally selects data for replay buffer by maximizing the diversity of samples in terms of the gradient in the parameter space. GEM (Lopez-Paz & Ranzato, 2017) and its variant A-GEM (Chaudhry et al., 2019) leverage an episodic memory that stores part of seen samples for each task to prevent forgetting old knowledge. Similarly, OGD (Farajtabar et al., 2020) stores gradients as opposed to actual data, providing a reference in projection. More recent work include GDumb (Prabhu et al., 2020), BiC (Wu et al., 2019), DER++, and Co$^2$L (Cha et al., 2021).

**General continual learning** Prior works on CL often rely on the information about the task boundaries. For example, some replay-based methods perform specific steps specifically at task boundaries, some regularization-based methods store network responses at these boundaries; architecture-based methods update the model architecture after one task is finished. However, when dealing with streaming data in practical settings, task boundaries are not clearly defined. This scenario, where sequential tasks are learned continuously without explicit knowledge of task boundaries, is referred to as *general continual learning* (Buzzega et al., 2020; Aljundi et al., 2019; Chaudhry et al., 2019). To address general continual learning, replay-based methods can utilize reservoir sampling (Vitter,

1985), which allows sampling throughout the training rather than relying on task boundaries. In our work, we specifically focus on general continual learning with reservoir sampling, particularly in the context of federated learning setups.

## E.2 FEDERATED LEARNING (FL)

There is a rapidly increasing level of interest in FL from both industry and academia, especially due to its benefits on enabling multiple users to collaboratively train a model with improved data privacy (Kairouz et al., 2021; Lim et al., 2020; Zhao et al., 2018; Konečný et al., 2016). FedAvg (McMahan et al., 2017) is a widely used algorithm in FL where each round consists of three steps: first, each client updates its local model using its data and transmits the updated local model to the server; second, the central server aggregates the updated local models and updates the global model in the direction of the average of local updates; third, the global model is broadcasted to each client and the local model is redefined as the global model; we repeat these three steps for multiple communication rounds. Variants of FedAvg were suggested in recent years (Li et al., 2020; Shoham et al., 2019; Karimireddy et al., 2020; Li et al., 2019; Mohri et al., 2019), but most existing works assume that the data distribution is static over time, which fails to capture the temporal dynamics of real-world data.

## E.3 CONTINUAL FEDERATED LEARNING (CFL)

CFL is a problem of learning multiple consecutive tasks in the FL setup. Several CFL methods have been proposed in the literature. FedProx (Li et al., 2020) adds a proximal term to limit the impact of variable local updates, while FedCurv (Shoham et al., 2019) adds a penalty term using the diagonal of the Fisher information matrix to protect the important parameters for each task. Both methods aim to preserve previously learned tasks while training new ones. Although FedProx (Li et al., 2020) and FedCurv (Shoham et al., 2019) are effective approaches to mitigate forgetting in CFL, they have been shown to achieve suboptimal performance when applied in a naïve manner.

Other approaches, such as FedWeIT (Yoon et al., 2021) and NetTailor (Venkatesha et al., 2022), have attempted to prevent interference between irrelevant tasks by decomposing network parameters or using a dynamic architecture approach. However, these methods may not be practical for a large number of tasks as the number of parameters scale linearly and require a clear understanding of all task identities or boundaries in advance, which may not be feasible in real-world scenarios. CFeD (Ma et al., 2022) and FedCL (Yao & Sun, 2020) utilize surrogate datasets or global importance weights to distill learned knowledge and constrain local model updates, respectively. However, these methods require extra effort to generate or collect auxiliary data and may consume extra communications or storage overhead. GLFC (Dong et al., 2022) is another approach that uses a class-aware gradient compensation loss and a class-semantic relation distillation loss to overcome catastrophic forgetting, but it only considers class-incremental learning scenarios.

Unlike previous methods, our approach does not require explicit task boundaries, making it more practical for real-world applications. Our method achieves this by aligning the gradients of the current model with those of the global buffer, which contains past experiences from multiple clients. By leveraging this collective experience, FedGP can effectively mitigate forgetting of previously learned knowledge in FL scenarios with continuous data and real-world temporal dynamics.

