# OpenReview forum: "FedGP: Buffer-based Gradient Projection for Continual Federated Learning"
_ICLR.cc/2024/Conference — Submitted to ICLR 2024_

### Official Review · Reviewer_sEUN · 2023-10-24

**Soundness:** 3 good
**Presentation:** 3 good
**Contribution:** 2 fair
**Rating:** 5
**Confidence:** 3

**Summary:**

The paper introduces a approach called FedGP for addressing the challenge of catastrophic forgetting in Continual Federated Learning, where decentralized clients learn from continuous data streams. FedGP aims to preserve knowledge across clients by leveraging local buffer samples and aggregated buffer gradients. This method enhances the performance of existing continual learning and CFL techniques and demonstrates performance improvements on standard benchmarks.

**Strengths:**

The experiment part is comprehensive and solid. The authors compare the proposed method with various baselines on a number of tasks. The experiment details and results are well presented and explained. The authors also report the storage, communication, computation overhead of the proposed methods, which is great importance to show the efficiency of the proposed method.

**Weaknesses:**

1. The novelty of the proposed method is questionable. The key steps of the proposed FedGP algorithm, including using buffers and performing gradient projection, can all be found in previous works [1]. The only major difference, as far as I can tell, is to extend the centralized setup in [1] to a decentralized/federated setup.
2. The algorithm design of FedGP is mostly heuristic and not well motivated. For example,  the authors do not rigorously explain why the gradient projection operation and the buffer updating algorithm will solve the catastrophic forgetting problem and  fit into the FL setting. A theoretical analysis and more ablations on algorithm design will help to show the effectiveness of FedGP.
3. I am skeptical about whether the use of a buffer can solve the problem of catastrophic forgetting. Since the buffer size is limited, one is unable to store the entire past information in it, and has to discard stale information. According to Algorithm 4, when the buffer is full the algorithm randomly selects an old entry in the buffer and replace it with a new data point. How can this avoid discarding useful information, e.g., the data point stored in previous rounds?

[1] Arslan Chaudhry, Marc’Aurelio Ranzato, Marcus Rohrbach, and Mohamed Elhoseiny. Efficient lifelong learning with A-GEM.

**Questions:**

See the weaknesses part

---

> ### Author Response · Authors · 2023-11-19
>
> We thank the Reviewer for the detailed review and constructive suggestions. We appreciate your acknowledgement that the experiment part is comprehensive, solid and well-written. Our responses are detailed below.
>
> **`Q1: The novelty of the proposed method is questionable. The key steps of the proposed FedGP algorithm, including using buffers and performing gradient projection, can all be found in previous works [1]. The only major difference, as far as I can tell, is to extend the centralized setup in [1] to a decentralized/federated setup.`**
>
> We consider our main contribution lies in the development and presentation of a ***simple yet highly effective*** algorithmic solution for continual federated learning. We have devoted considerable effort to an **extensive empirical validation** of our approach. Especially, to support our claim, we added new extensive experimental results in the revised manuscript. The detailed explanations can be found in a related question [R1-Q1] by Reviewer MNXy.
>
> [1] Efficient lifelong learning with A-GEM.
>
>
>
> **`Q2: The algorithm design of FedGP is mostly heuristic and not well motivated. For example, the authors do not rigorously explain why the gradient projection operation and the buffer updating algorithm will solve the catastrophic forgetting problem and fit into the FL setting. A theoretical analysis and more ablations on algorithm design will help to show the effectiveness of FedGP.`**
>
>
> Thanks for the sharp question. To address the reviewer's concern, below we added extensive experimental results for ablations on our algorithm design, which is decomposed into (1) gradient manipulation algorithm, and (2) buffer updating algorithm.
>
> First, we considered different ways of manipulating the gradient $g$, given the reference gradient $g_{\text{ref}}$. In the below table, we compared **four different methods of updating $g$**:
> - ``Average``: define $g \leftarrow (g+g_{\text{ref}})/2$.
> - ``Rotate`` : rotate $g$ towards $g_{\text{ref}}$ while keeping the magnitude.
> - ``Project``: project $g$ to the space that is orthogonal to $g_{\text{ref}}$.
> - ``Project & Scale``: apply ``Project`` and then scale up the vector such that the magnitude is identical to the original $g$
>
> Recall that our FedGP applies ``Project`` method only when the angle between $g$ and $g_\text{ref}$ is larger than 90 degree, i.e., when the reference gradient $g_\text{ref}$ (measured for the previous tasks) and the gradient $g$ (measured for the current task) conflicts to each other. Our intuition for such choice is, it is better to manipulate $g$ if the direction favorable for current task is conflicting with the direction  favorable for previous tasks. To support that this choice is meaningful, we compared **two ways of deciding when we manipulate the gradients**, denoted below:
>
> - ``(>90)``: update the gradient $g$ only when the angle($g,g_\text{ref}$)>90
> - ``(Always)``: update the gradient $g$ always
>
>
> We compared the performances of above choices in Table [R3-2a], for S-CIFAR100 dataset. One can confirm that our FedGP (denoted by ``Project (>90)`` in the table) far outperforms all other combinations, **showing that our design (doing projection for conflicting case only) is the right choice**. If we check each component (``Project`` and ``(>90)``) independently, one can check that choosing ``Project`` outperforms ``Average``, ``Rotate`` and ``Project & Scale`` in most cases, and choosing  ``(>90)`` outperforms ``Always`` in all cases.
>
>
> #### [Table R3-2a. Effect of different gradient manipulation method on the accuracy (%) of FedGP, tested on S-CIFAR100]
> |Method|Class-IL|Task-IL|
> |-|-|-|
> |FL|8.76±0.1|47.74±1.2|
> |Average (Always)|7.26±1.95|35.96±3.23|
> |Average (>90)|7.79±0.65|36.57±1.55|
> |Rotate (Always)|7.59±0.89|36.15±2.83|
> |Rotate (>90)|8.41±0.78|38.97±1.83|
> |Project \& Scale (Always)|8.77±0.09|32.96±1.10|
> |Project \& Scale (>90)|12.30±0.65|73.61±0.75|
> |Project (Always)|8.90±0.08|34.00±1.98|
> |Project (>90), **ours**|**17.08**±1.8|**74.71**±0.9|
>
>
> We also tested whether doing the projection is helpful in all cases when $\text{angle}(g, g_\text{ref}) > 90$. We considered applying the projection for $p\%$ of the cases having $\text{angle}(g, g_\text{ref}) > 90$, for $p=10, 50, 80$ and $100$. Note that $p=100\%$  case reduces to our FedGP.
>
> Table R3-2b shows the effect of **projection rate** $p\%$ on the accuracy, tested on S-CIFAR100 dataset. In both class-IL and task-IL settings, increasing $p$ always improves the accuracy of the FedGP method. This supports that our way of projection is suitable for the continual federated learning setup.
>
>
> #### [Table R3-2b. Effect of projection rate $p\%$ on the accuracy (%) of FedGP, tested on S-CIFAR100]
> |Method|Class-IL|Task-IL|
> |-|-|-|
> |FL, $p=0\%$|8.76±0.1|47.74±1.2|
> |FedGP, $p=10\%$|8.82±0.07|54.90±1.61|
> |FedGP, $p=50\%$|8.91±0.07|67.89±0.67|
> |FedGP, $p=80\%$ |10.36±0.42|72.73±0.74|
> |FedGP, $p=100\%$ (Ours)|**17.08**±1.8|**74.71**±0.9|

---

> ### Author Response · Authors · 2023-11-19
>
> In Table R3-2c, we compared three different **buffer updating algorithms**:
> - ``Random Sampling``: randomly replaces a data point in the buffer with incoming new data
> - ``Sliding Window Sampling``: replaces the earliest data point in the buffer when new data arrives
> - ``Reservoir Sampling`` (Ours): given $N$ (the number of observed samples up to now) and $B$ (the buffer size), we do the following
>     - when $N \le B$, we put the current sample in the buffer
>     - when $N > B$, with probability $B/N < 1$, we replace a random sample in the buffer with the current sample
>
> Note that when the number of incoming data is $N$, those $N$ samples have the same probability of getting into the buffer, for the ``Reservoir Sampling`` method used in our paper. Thus, when ``Reservoir Sampling`` is used, the **buffer contains approximately equal number of samples for each task** (when each task has the same number of samples), throughout the continual learning process. This is the underlying intuition why we choose such buffer updating algorithm. To support this claim, we report the performance of different sampling methods in Table R3-2c. Here, one can confirm that our sampling method is outperforming other sampling methods.
>
>
> #### [Table R3-2c. Effect of different buffer updating algorithms on the accuracy (%) of FedGP, tested on CIFAR100]
> |Method|Class-IL|Task-IL|
> |-|-|-|
> |FL|8.76±0.1|47.74±1.2|
> |Sliding Window Sampling|8.82±0.15|46.16±2.38|
> |Random Sampling|9.72±0.10|54.82±1.58|
> |Reservoir Sampling **(Ours)**|**17.08**±1.8|**74.71**±0.9|
>
>
>
>
>
> The below figure compares the sample distribution across different tasks, for different buffer updating algorithms (uniform-random sampling, reservoir sampling and sliding window sampling). This clearly shows the difference of buffer updating algorithms, which guided to the different performances reported in Table R3-2c.
>
> https://docs.google.com/document/d/e/2PACX-1vRzMZ8Ha3DaG79IhAR6lLEFj7UYjQt-W3pJkBvgkPHLtSIv5KkCE7DcAUIBf7VuxnUVM3l78oduC54N/pub
>
> All in all, the reservoir sampling method used in FedGP allows us to have **balanced sample distribution across different tasks**, thus allowing us to mitigate the catastrophic forgetting and to improve the accuracy in the continual federated learning setting.
>
> In the revised manuscript, we will include our new results as well as the discussion we made during the rebuttal.
>
>
>
>
>
> **`Q3: I am skeptical about whether the use of a buffer can solve the problem of catastrophic forgetting. Since the buffer size is limited, one is unable to store the entire past information in it, and has to discard stale information. According to Algorithm 4, when the buffer is full the algorithm randomly selects an old entry in the buffer and replace it with a new data point. How can this avoid discarding useful information, e.g., the data point stored in previous rounds?`**
>
> Thanks for the great question. The reviewer is correct that a finite buffer can**not** maintain the entire history of data regardless of buffer update algorithm. Instead, our goal is to continuously maintain the buffer so that it represents all the tasks that have been seen in the past. Under the ***Reservoir Sampling***, it is easy to prove that after seeing $N$ samples for any $N$, the samples stored in the buffer are $B$ samples chosen uniformly at random. Before proving this property, let us recall the ***definition of Reservoir Sampling***: for a given $N$ (the number of observed samples up to now) and $B$ (the buffer size),
> - when $N \le B$, we put the current sample in the buffer
> - when $N > B$, with probability $B/N < 1$, we replace a random sample in the buffer with the current sample
>
>
> We now prove that for reservoir sampling, the probability of a sample contained in the buffer is $\frac{B}{N}$. We prove this by induction; suppose this statement holds when $N-1$ samples are observed, and we show that this holds when one more sample is observed. Note that the probability of a sample contained in the buffer can be computed as $\frac{B}{N-1} * (1-\frac{B}{N}*\frac{1}{B})=\frac{B}{N}$, where
>
> - $\frac{B}{N-1}$ is the probability of the sample initially contained in the buffer;
> - $(1-\frac{B}{N}*\frac{1}{B})$ is the probability of a sample not being kicked out of the buffer;
> - $\frac{B}{N}*\frac{1}{B}$ is the probability of a sample being kicked out of the buffer
>
> We will include these details in the revised manuscript.

---

> ### Author Response · Authors · 2023-11-19
>
> From the above mathematical results, we can confirm that the reservoir sampling allows the buffer to contain samples from previous tasks. However, as the reviewer mentioned, given a limited buffer size, the number of samples the buffer can maintain for old tasks is bounded above. In Fig. R3-3, we reported the effect of buffer size on the accuracy of the trained model for old tasks. At the end of each task, we measured the accuracy of the trained model with respect to the test data for task 1. We tested on S-CIFAR100 dataset, and considered task incremental learning (task-IL) setup.
>
> One can observe that when the buffer size $B$ is small, the accuracy drops as the model is trained on new tasks. On the other hand, when $B \ge 100$, the task-IL accuracy for task 1 is maintained throughout the process. Note that training with our default setting $B=200$ does not hurt the accuracy for task 1 throughout the continual learning process.
>
>
> #### [Figure R3-3. Accuracy (%) for Task 1, under the Task-IL setting on S-CIFAR100. We tested on FedGP with different buffer sizes.]
>
> https://docs.google.com/document/d/e/2PACX-1vQfbETzt5SUSDrgcCAypbjrxTia1uBgTjNZ4_KnY6tYZ7pcpRo20SA-FiDctxgQNKUwTgP_RZSRxgdZ/pub

---

### Official Review · Reviewer_Nmd1 · 2023-10-31

**Soundness:** 3 good
**Presentation:** 3 good
**Contribution:** 3 good
**Rating:** 5
**Confidence:** 4

**Summary:**

This paper introduces FedGP, a buffer-based gradient projection method designed for continual federated learning. FedGP effectively tackles the challenge of catastrophic forgetting while amplifying the performance of the continual learning and continual federated learning techniques. Extensive experiments conducted on multiple benchmarks demonstrate the effectiveness of the proposed FedGP method.

**Strengths:**

1. This paper introduces a buffer-based gradient projection method to address the catastrophic forgetting in continual federated learning.
2. This work is straightforward and easily understandable, making it reader-friendly for potential readers.
3. Extensive experimental results demonstrated the effectiveness of the proposed FedGP method.

**Weaknesses:**

The paper focus on an interesting continual federated learning problem and has several issues that can be improved: The proposed FedGP method suffer from more communication costs than comparison methods. Moreover, additional gradients need to be uploaded to the server which increases the risk of the local data leakage.

**Questions:**

(1) The gradient projection that conflict with the previous task update direction would aggravate forgetting, while directly drop it would impair the current task learning. The proposed method can be addressed the forgetting to some extent while ignoring its impact on the current task.

(2) In Table 3, the two times communication get an inferior performance than the equalized communication with FedGP in P-MNIST, can you give an explanation?

(3) The adopted comparison methods seem to be outdated, it is better to adopted the SOTA methods to further validate the effectiveness of the proposed FedGP method.

(4) The literature[1] also addressed the forgetting by using gradient projection, which has a strong connection with the proposed method. It is better to taken as a comparison method.

(5) Please see the weakness above.

[1] Gradient Projection Memory for Continual Learning, In ICLR 2021

---

> ### Author Response · Authors · 2023-11-19
>
> We thank Reviewer Nmd1 for the detailed review and very helpful suggestions. We thank the Reviewer for appreciating the proposed projection method, extensive experimental results, and our efforts in making a well-written paper. Our responses are detailed below.
>
> **`Q1: The gradient projection that conflict with the previous task update direction would aggravate forgetting, while directly drop it would impair the current task learning. The proposed method can be addressed the forgetting to some extent while ignoring its impact on the current task.`**
>
>
> We thank the Reviewer for pointing this out. To address your concern, below we provided additional experimental results on the performance measured for the **current** task. The below plot shows the Class-IL accuracy of FedGP (with buffer size 200) and FL for S-CIFAR100, where the total number of tasks is set to 10. During the continual learning process, we measured the accuracy of each model for the **current** task. One can confirm that using FedGP does not hurt the current task accuracy, compared with FL. Note that this shows that FedGP does not impair the performance of the current task, while also alleviating the forgetting in upcoming rounds.
>
>
> We will include this result and discussion in the revised manuscript.
>
>
>
>
> #### [Figure R2-1. Class-IL Accuracy (%) of current task for FL and FedGP on the S-CIFAR100]
>
> https://docs.google.com/document/d/e/2PACX-1vSqmIgnlzdcX2Pvvjw3acXsM1tINU-Jj385cYadSk1jUI9RDm6jr7hZQ3mtomf3Mxo6nDYBq4ZR9Lre/pub
>
>
>
> **`Q2: In Table 3, the two times communication get an inferior performance than the equalized communication with FedGP in P-MNIST, can you give an explanation?`**
>
> We thank the Reviewer for carefully reading our work.
> After reading our result again, we noticed that the performances of two methods (``2x communication`` stragety and ``equalized communication`` strategy) in P-MNIST are quite close to each other, and they are within the errors.
>
>
> During the rebuttal period, we ran additional experiments; instead of the initial 5 runs (which we reported in the submitted manuscript), we did 20 runs per method, each with a unique random seed, the result of which is given below.
>
>
> #### [Table R2-2. Effect of communication on accuracy (%) performance averaged across 20 runs, for P-MNIST]
> |Method|Domain-IL|
> |-|-|
> |FL|27.49±1.98|
> |FL w/ FedGP (2× comm overhead) |**35.91**±3.98|
> |FL w/ FedGP (equalized comm overhead) |34.96±3.16|
>
>
> As in Table R2-2, ``2x communication`` strategy has a slightly better performance than ``equalized communication`` strategy, when evaluated across 20 runs. Note that this result is consistent with our finding for other datasets (R-MNIST, S-CIFAR10 and S-CIFAR100) we reported in Table 3 of the submitted manuscript.
>
> Based on this observation, we will revise the result for P-MNIST in the updated manuscript.

---

> ### Author Response · Authors · 2023-11-19
>
> **`Q3: The adopted comparison methods seem to be outdated, it is better to adopted the SOTA methods to further validate the effectiveness of the proposed FedGP method.`**
>
> As per the reviewer's suggestion, we compared our method with SOTA paper [1] proposing Federated Orthogonal Training (FOT) algorithm for continual federated learning.
>
> In FOT, at the end of each task, each local client transmits the activation vectors information (computed for local data points) of each local model to the server. Then, the server computes the subspace $S$ spanned by the aggregated activations. This subspace is used during the global model update process; the gradient $g$ is updated in the direction that is orthogonal to the subspace $S$. Note that FOT has several advantages compared with existing baselines; the privacy leakage is mitigated, the communication cost is reduced, and the solution has theoretical guarantees. We very much appreciate their work.
>
> While both FOT and our FedGP project the gradients $g$ on the subspace $S$ specified by previous tasks, they have two main differences. First, the subspace $S$ is defined in a different manner: FOT relies on the representations of local model activations to define $S$. FedGP, on the other hand, relies on the gradient of model computed on its local buffer data. Second, FOT projects the gradient computed at the server side, while FedGP projects the gradient computed at each local client.
>
> The below table compares the accuracy of FL, FOT and FedGP (with buffer size 200) for P-MNIST and S-CIFAR100 under the task incremental learning scenario, consistent with the paper's benchmarks. Assuming that the local buffer is available, FedGP outperforms FOT.
>
> We will include this in the revised manuscript.
>
> #### [Table R2-3.  Comparative accuracy (%) performance analysis of FL, FOT, and FL+FedGP]
>
> | Methods  | P-MNIST (Domain-IL) | S-CIFAR100 (Task-IL) |
> |----------|---------------------|----------------------|
> | FL       | 25.92±2.1           | 47.74±1.2            |
> | FOT      | 23.77±1.1           | 50.57±1.5            |
> | FL+FedGP | **34.23**±2.7           | **74.71**±0.9            |
>
> [1] "Federated Orthogonal Training: Mitigating Global Catastrophic Forgetting in Continual Federated Learning"
>
> **`Q4: The literature[1] also addressed the forgetting by using gradient projection, which has a strong connection with the proposed method. It is better to taken as a comparison method.`**
>
> We thank the Reviewer for bringing this work [1] to our attention. As the reviewer mentioned, both our method (FedGP) and the gradient projection memory (GPM) method suggested in [1] consider gradient projection for mitigating the forgetting issue.
>
> However, FedGP (ours) and GPM (proposed in [1]) have two clear differences.
>
> First, the reference vector used for the projection is different. In our FedGP, the reference vector is the gradient $g_{\text{ref}}$ of the global model computed by the buffer data stored at clients. In contrast, in [1], the network representation (activations) approximated by top singular vectors are used as the reference vector, which is measured when each task ends.
>
> Second, GPM requires task boundaries, while FedGP does not. Thus, FedGP can be applied in more realistic scenarios where the task boundaries are not given.
>
> Table R2-4 compares the performance of FedGP and GPM (applied in federated learning) for various datasets. We compared the two methods assuming that task boundaries are given. Note that our method does not use this information, while GPM does. Still, we observe that our FedGP outperforms GPM. The potential reasons could be (1) FedGP utilizes buffer gradients which could provide more information than the top singular vectors derived from the activations in GPM. (2) FedGP updates its reference gradients more often, like after every communication round, while GPM only updates after finishing a task.
>
> [1] “Gradient projection memory for continual learning.”
>
> #### [Table R2-4. Comparative accuracy (%) performance analysis of FL, FL+GPM, and FL+FedGP across various datasets]
> | Methods       | R-MNIST (Domain-IL) | S-CIFAR10 (Class-IL) | S-CIFAR10 (Task-IL) |
> |---------------|---------------------|----------------------|---------------------|
> | FL            | 68.02±3.1           | 17.44±1.3            | 70.58±4.0           |
> | FL+GPM  | 74.42±6.4           | 17.59±0.4            | 74.50±3.6           |
> | FL+FedGP      | **79.46**±4.1           | **18.02**±0.6           | **80.83**±2.0           |
>
> |          | P-MNIST (Domain-IL) | S-CIFAR100 (Class-IL) | S-CIFAR100 (Task-IL) |
> | -------- | ------------------- | --------------------- | -------------------- |
> | FL       | 25.92±2.1           | 8.76±0.1              | 47.74±1.2            |
> | FL+GPM   | 31.92±3.4           | 8.18±0.1              | 54.48±1.4            |
> | FL+FedGP | **34.23**±2.7                        | **17.08**±1.8             |**74.71**±0.9
>
> We will include this in the revised manuscript.

---

> ### Author Response · Authors · 2023-11-19
>
> **`Q5: The paper focus on an interesting continual federated learning problem and has several issues that can be improved: The proposed FedGP method suffer from more communication costs than comparison methods. Moreover, additional gradients need to be uploaded to the server which increases the risk of the local data leakage.`**
>
> We thank the reviewer for pointing out the possible limitations of our method. Regarding the communication cost, it is true that vanilla FedGP has 2x communication cost compared with baselines. However, we consider a variant of FedGP which updates the model and buffer gradient less frequently (i.e., reduce the communication rounds for each task by half or more), which has reduced communication than the vanilla FedGP.
>
>
> During the rebuttal period, we ran extensive new experiments which explore the accuracy of FedGP methods for different communication frequencies. Table R2-5 below reports the performance for different datasets, when the communication overhead is set to 2x, 1x, 0.5x and 0.2x. Values in brackets indicate differences from the FL baseline.
>
> First, in most cases (except S-CIFAR10 Class-IL setup), FedGP with equalized (1x) communication overhead is outperforming FL. In addition, for most of tested datasets including R-MNIST, P-MNIST and S-CIFAR100, FedGP outperforms FL with at most 0.5x communication overhead. This means that FedGP enjoys a **higher performance with less communication**, in the standard benchmark datasets for continual federated learning.
>
> In the revised manuscript, we will include Table R2-5 our discussions written above.
>
>
>
> #### [Table R2-5. Effect of communication on the accuracy (%) performance, with values in brackets indicating differences from the FL baseline]
>
> Here's the updated table with the differences compared to the first row (FL) included:
>
> | Method                                   | R-MNIST                  | P-MNIST                 | S-CIFAR10 Class-IL       | S-CIFAR10 Task-IL       | S-CIFAR100 Class-IL     | S-CIFAR100 Task-IL     |
> | ---------------------------------------- | ------------------------ | ----------------------- | ------------------------ | ----------------------- | ----------------------- | ---------------------- |
> | FL                                       | 68.02±3.1                | 27.49±1.98              | 17.44±1.3                | 70.58±4.0               | 8.76±0.1                | 47.74±1.2              |
> | FL w/ FedGP (2x comm overhead)           | **79.46**±4.1 (↑11.44)   | **35.91**±3.98 (↑8.42)  | **18.02**±0.6 (↑0.58)   | **80.83**±2.0 (↑10.25)  | **17.08**±1.8 (↑8.32)   | **74.71**±0.9 (↑26.97) |
> | FL w/ FedGP (equalized comm overhead)    | 75.63±3.9 (↑7.61)        | 34.96±3.16 (↑7.47)      | 16.65±1.0 (↓0.79)       | 78.79±2.8 (↑8.21)       | 13.62±0.6 (↑4.86)       | 73.96±0.4 (↑26.22)     |
> | FL w/ FedGP (0.5x comm overhead)         | 76.05±4.05 (↑8.03)       | 29.75±4.63 (↑2.26)      | 14.30±1.34 (↓3.14)      | 66.90±3.61 (↓3.68)      | 13.09±0.49 (↑4.33)      | 69.96±0.56 (↑22.22)    |
> | FL w/ FedGP (0.2x comm overhead)         | 70.59±4.70 (↑2.57)       | 15.51±2.71 (↓11.98)     | 13.37±2.64 (↓4.07)      | 59.75±6.36 (↓10.83)     | 13.59±0.87 (↑4.83)      | 59.31±1.59 (↑11.57)    |
>
>
> Regarding the data leakage issue, we are running experiments measuring the privacy leakage of FedGP and other baselines. We will report the result as soon as we get it.

---

> > ### Author Response · Authors · 2023-11-23
> >
> > We thank the reviewer again for pointing out the possible limitations of our method.
> >
> > Regarding the data leakage issue, we are currently engaged in comprehensive experiments. Our focus is on empirically evaluating the privacy leakage of the FedGP framework and various other CFL baselines. For this purpose, we are employing the Mutual Information Neural Estimator (MINE), as outlined in [1]. This approach allows us to estimate the mutual information between individual user model updates and the aggregated model update across all users at each global training round. This methodology closely follows the precedent set in [2].
> >
> > We anticipate that these experiments will yield insightful results. We will include these results and relevant discussions if we have chances to revise the manuscript for camera ready.
> >
> > [1] Belghazi, Mohamed Ishmael, et al. "Mutual information neural estimation." International conference on machine learning. PMLR, 2018.
> >
> > [2] Elkordy, Ahmed Roushdy, et al. "How much privacy does federated learning with secure aggregation guarantee?." arXiv preprint arXiv:2208.02304 (2022).

---

### Official Review · Reviewer_MNXy · 2023-11-01

**Soundness:** 3 good
**Presentation:** 3 good
**Contribution:** 2 fair
**Rating:** 5
**Confidence:** 3

**Summary:**

This paper presents FedGP, an approach that addresses the challenges of continual federated learning (CFL) by integrating techniques from both continual learning (CL) and federated learning (FL). At its core, FedGP conducts server aggregation of local buffer gradient to obtain a global buffer gradient. Then use the global buffer gradient to guide the local continual learning by gradient projection. Furthermore, in the federated context, FedGP preserves the knowledge across multiple tasks in a  non-iid and heterogeneous client data distribution. Through this integration, FedGP improves the performance of current CFL approaches.

**Strengths:**

The paper is easy to read and follow, and the extensive experiments show the resulting improvement spanning many datasets and baseline methods. The proposed methods also work with a large number of clients and partial client aggregation in FL.

**Weaknesses:**

1. While I'm familiar with FL and CL, I'm not a deep expert in CFL. Observing FedGP, it seems to be a federated take on A-GEM, where g_ref is replaced by a federated aggregation of local buffer gradients. Although FedGP shows promising results when combined with other methods, I'm still pondering its overall novelty and contribution.

2. In the experimental section, Table 1 presents the performance of all baseline methods. It's noticeable that, in the without FedGP column, even though CFL methods are designed for continual learning challenges, some, especially FL+iCaRL/L2P, performed better than CFL methods. Can the authors clarify the reasons behind these results?

**Questions:**

See weakness.

---

> ### Author Response · Authors · 2023-11-19
>
> We thank Reviewer MNXy for the detailed review. We are happy to see the acknowledgement of the paper's clarity and thorough experiments. Please find our answers to comments and questions as follows.
>
> **`Q1: Observing FedGP, it seems to be a federated take on A-GEM, where g_ref is replaced by a federated aggregation of local buffer gradients. Although FedGP shows promising results when combined with other methods, I'm still pondering its overall novelty and contribution.`**
>
>
>
> Thanks for your comments.
>
> We consider our main contribution lies in the development and presentation of a ***simple yet highly effective*** algorithmic solution for continual federated learning. Our FedGP is a simple variant combining existing methods, but it highly outperforms baselines in various scenarios. To show this, we have devoted considerable effort to an **extensive empirical validation** of our approach, including
> * image classification
> * language processing
> * object detection data from the CARLA simulator
>
> This empirical justification is a critical part of our contribution, demonstrating the general applicability of our algorithm.
>
> To further strengthen our claim, we **added extensive new results during the rebuttal period**. To be specific, we conducted experiments exploring the effect of
> * gradient manipulation (projection, rotation, averaging) methods -- result is in R3-Q2
> * buffer data sampling methods (reservior, random, sliding-window)-- result is in R3-Q2
> * buffer size -- result is in R3-Q3
> * communication frequency -- result is in R2-Q5
> * the reference gradient used for projection (by comparing with related baselines [1], [2]) -- result is in R2-Q3 & R2-Q4
>
>
> [1] Saha, Gobinda, Isha Garg, and Kaushik Roy. "Gradient projection memory for continual learning." arXiv preprint arXiv:2103.09762 (2021).
> [2] Bakman, Yavuz Faruk, et al. "Federated Orthogonal Training: Mitigating Global Catastrophic Forgetting in Continual Federated Learning." arXiv preprint arXiv:2309.01289 (2023).
>
>
>
>
>
> **`Q2: In the experimental section, Table 1 presents the performance of all baseline methods. It's noticeable that, in the without FedGP column, even though CFL methods are designed for continual learning challenges, some, especially FL+iCaRL/L2P, performed better than CFL methods. Can the authors clarify the reasons behind these results?`**
>
>
> We thank the reviewer for the sharp question.
> First, it is worth nothing that **FL+CL methods could still outperform some CFL methods under certain conditons**, and the similar findings have been made in the previous work too. For example, according to a recent work [1], FL+CL methods (FL+EWC and FL+ER) outperforms CFL methods (GLFC and FedCIL) for S-CIFAR100 dataset.
>
> Second, we note that FL+L2P's high performance is due to a different reason -- **it utilizes a pretranied Vision Transformer (ViT), which helps mitigate the catastrophic forgetting**. This is why we wrote the numbers in gray with a caveat in the caption, but we will further clarify this in the revision.
>
>
>
>
> Third, regarding a CFL method called CFeD [2], it is important to note that its performance is heavily dependent on the choice of surrogate data (we used CIFAR for our experiments), and **no direct comparison with state-of-the-art FL+CL methods (e.g., iCaRL) has been reported** in [2].
>
>
>
> In the revised manuscript, we will include this discussion.
>
> [1] Y. F. Bakman et al., "Federated Orthogonal Training: Mitigating Global Catastrophic Forgetting in Continual Federated Learning", arXiv 2023
> [2] Ma, Yuhang, et al. "Continual federated learning based on knowledge distillation." Proceedings of the Thirty-First International Joint Conference on Artificial Intelligence. Vol. 3. 2022.

---

### Author Response · Authors · 2023-11-19

**`Q2. (paraphrased) Comparison with Continual (Federated) Learning methods, such as GPM or other State-of-the-Art (SOTA) techniques.`**


We added experiments comparing our method with recent baselines (GPM [1] and FOT [2] uploaded in 2023). GMP and FOT utilize principal subspace of network activations at the end of each task, offering advantages such as reduced memory and enhanced privacy. Furthermore, FOT gives theoretical guarantees in their work. We very much appreciate their work.

Below tables show that FedGP outperforms these baselines throughout different datasets we tested. In the revised manuscript, we will cite [1] and [2], clearly state the difference of FedGP and those baselines, and include the comparison table. The detailed explaination of each method and comparison of FedGP and baselines are given in our response for Q3 and Q4 of Reviewer Nmd1. Please refer to our detailed reponse in the following.




[1] Saha, Gobinda, Isha Garg, and Kaushik Roy. “Gradient projection memory for continual learning.” arXiv preprint arXiv:2103.09762 (2021).
[2] Y. F. Bakman et al., "Federated Orthogonal Training: Mitigating Global Catastrophic Forgetting in Continual Federated Learning", arXiv 2023


#### [Table R2-3.  Comparative accuracy (%) performance analysis of FL, FOT, and FL+FedGP]

| Methods  | P-MNIST (Domain-IL) | S-CIFAR100 (Task-IL) |
|----------|---------------------|----------------------|
| FL       | 25.92±2.1           | 47.74±1.2            |
| FOT      | 23.77±1.1           | 50.57±1.5            |
| FL+FedGP | **34.23**±2.7           | **74.71**±0.9            |




#### [Table R2-4. Comparative accuracy (%) performance analysis of FL, FL+GPM, and FL+FedGP across various datasets]
| Methods       | R-MNIST (Domain-IL) | S-CIFAR10 (Class-IL) | S-CIFAR10 (Task-IL) |
|---------------|---------------------|----------------------|---------------------|
| FL            | 68.02±3.1           | 17.44±1.3            | 70.58±4.0           |
| FL+GPM  | 74.42±6.4           | 17.59±0.4            | 74.50±3.6           |
| FL+FedGP      | **79.46**±4.1           | **18.02**±0.6           | **80.83**±2.0           |

|          | P-MNIST (Domain-IL) | S-CIFAR100 (Class-IL) | S-CIFAR100 (Task-IL) |
| -------- | ------------------- | --------------------- | -------------------- |
| FL       | 25.92±2.1           | 8.76±0.1              | 47.74±1.2            |
| FL+GPM   | 31.92±3.4           | 8.18±0.1              | 54.48±1.4            |
| FL+FedGP | **34.23**±2.7                        | **17.08**±1.8             |**74.71**±0.9



------



**`Q3. (paraphrased) Novelty and contribution of our paper.`**

We consider our main contribution lies in the development and presentation of a ***simple yet highly effective*** algorithmic solution for continual federated learning. We have devoted considerable effort to an **extensive empirical validation** of our approach. Especially, to support our claim, we added new extensive experimental results in the rebuttal document (see our response to Q1 in AC and all reviewers section). The detailed explanations can be found in our response to Q1 by Reviewer MNXy.

---

### Author Response · Authors · 2023-11-19

# To AC and all Reviewers

We thank the reviewers for their insightful feedback and constructive comments and for providing suggestions to improve our paper.

First, we are encouraged by the positive feedback from the reviewers. Key highlights include:

- Extensive, comprehensive and solid experimental parts (**R-MNXy**, **R-Nmd1**, **R-sEUN**) demonstrated the effectiveness of the proposed FedGP method.
- The FedGP method is efficient (**R-sEUN**) and can be applied in various scenarios (**R-MNXy**).
- The paper is well written, and the FedGP method is easy to understand (**R-Nmd1**, **R-sEUN**).

As for the concerns/questions raised, we believe that we successfully addressed every single one, as replied to each reviewer. The major comments are summarized below.

**`Q1. (paraphrased) What is the motivation of the algorithm design (e.g, manipulating gradients and updating buffer data) of FedGP? Why did we choose this design, and how does it work effectively?`**

**First, we added explanation on why our (1) gradient projection method and (2) buffer updating algorithm are suitable for the continual learning setting**.

Recall that FedGP projects the gradinet $g$ on the subspace that is orthogonal to the reference gradient $g_\text{ref}$, only when the angle between $g$ and $g_\text{ref}$ is larger than 90 degree. In other words, we manipulate the gradient $g$ only when the reference gradient $g_\text{ref}$ (measured for the previous tasks) and the gradient $g$ (measured for the current task) conflicts to each other. Our intuition for such choice is, it is better to manipulate $g$ if the direction favorable for current task is *conflicting* with the direction favorable for previous tasks.

In addition, recall that FedGP uses reservoir sampling for updating the buffer data, which does the following: given $N$ (the number of observed samples up to now) and $B$ (the buffer size),
- when $N \le B$, we put the current sample in the buffer
- when $N > B$, with probability $B/N < 1$, we replace a random sample in the buffer with the current sample

It can be proved that when the number of incoming data is $N$, those $N$ samples have the same probability of getting into the buffer, for the reservoir sampling method used in FedGP. Thus, the **buffer contains approximately equal number of samples for each task** (when each task has the same number of samples), throughout the continual learning process. This is the underlying intuition why we choose such buffer updating algorithm.


Please check our response to Q2 of Reviewer sEUN for the further details.




**Second, we added extensive experimental results for ablations on our algorithm design.** The details of the below response is given in our response to Q2 of Reviewer sEUN.

Regarding the gradient manipulation algorithm, we tested on 8 different methods that use the reference gradient $g_{\text{ref}}$ to manipulate the gradient $g$, details of which are provided in our response to Q2 of Reviewer sEUN.

As in Table [R3-2a] below, our FedGP (denoted by ``Project (>90)`` in the table) far outperforms all other combinations, showing that our design choice (doing projection for conflicting case only) is the right choice.


#### [Table R3-2a. Effect of different gradient manipulation method on the accuracy (%) of FedGP, tested on S-CIFAR100]
|Method|Class-IL|Task-IL|
|-|-|-|
|FL|8.76±0.1|47.74±1.2|
|Average (Always)|7.26±1.95|35.96±3.23|
|Average (>90)|7.79±0.65|36.57±1.55|
|Rotate (Always)|7.59±0.89|36.15±2.83|
|Rotate (>90)|8.41±0.78|38.97±1.83|
|Project \& Scale (Always)|8.77±0.09|32.96±1.10|
|Project \& Scale (>90)|12.30±0.65|73.61±0.75|
|Project (Always)|8.90±0.08|34.00±1.98|
|Project (>90), **ours**|**17.08**±1.8|**74.71**±0.9|




Regarding the buffer updating algorithm, we compared three different methods (which are explained in our response to Q2 of Reviewer sEUN), as in Table R3-2c below. One can confirm that our reservoir sampling far outperforms other baselines.


#### [Table R3-2c. Effect of different buffer updating algorithms on the accuracy (%) of FedGP, tested on S-CIFAR100]
|Method|Class-IL|Task-IL|
|-|-|-|
|FL|8.76±0.1|47.74±1.2|
|Sliding Window Sampling|8.82±0.15|46.16±2.38|
|Random Sampling|9.72±0.10|54.82±1.58|
|Reservoir Sampling **(Ours)**|**17.08**±1.8|**74.71**±0.9|


Please check our response to Q2 of Reviewer sEUN for the further details.







------

---

### Author Response · Authors · 2023-11-21

Dear Reviewers, we have included additional results and discussions in the revised manuscript (highlighted in blue), based on your valuable feedback. We have also made several improvements to enhance the paper's clarity. Thanks again for your great comments and we would love to hear any further thoughts you may have.

---

### Meta-Review · Area_Chair_MVq9 · 2023-12-22

**Metareview:**

In its initial version, the paper's main contribution was characterized as introducing a buffer-based gradient projection method (FedGP), as a novel method to mitigate catastrophic interference across multiple tasks in continual federated learning. The previous sentence puts emphasis on the words "initial version", as it seems through reviews, rebuttal and proposed changes, this focus is being shifted to, quoting the authors,  "presentation of a simple yet highly effective algorithmic solution for continual federated learning" and "extensive empirical evaluation". Overall, there is a large amount of content that has been added in the rebuttal, partially modified in the paper, and suggested to be included in a final version.

The decision for the paper was by no means easy. All reviewers gave initial borderline ratings and shared some common concerns, in addition to their individual takes. Primarily, a large concern has revolved around the potential lack of novelty of the proposed new method, FedGP, which going beyond mere inspiration, seems largely an adoption of well-known existing techniques on orthogonal gradients from continual learning to the joint federated set-up. As such, some reviewers have expressed that additional theoretical and empirical insight is required to justify why this method is particularly suitable beyond the perhaps likely obvious insight that auxiliary memory buffers provide boosts to performance in almost all applications (taking into account that they do not explicitly interfere). As stated above through the quotes of the authors, the rebuttal seems to have focused primarily on the addition of empirical evaluation, to corroborate the experimental utility and nuance application.

Ultimately, given the lack of ongoing discussion post rebuttal with all reviewers, the AC has spent significant amount of time to go through the paper in detail, and to analyze whether the different reviewers' concerns were indeed addressed by the rebuttal. In this process, the AC has also spent time on reading and comparing to a very recently proposed related paper - abbreviated as federated orthogonal training (FOT) in the added comparison of the authors' rebuttal. At the end of this process, the AC has decided to suggest that the paper be rejected in its present form. This decision is in spite of being aware of both the fact that the empirical comparison outperforms the recent theoretically more motivated contender FOT, as well as the many suggested additions on experiments indeed addressing partial concerns of the reviewers. In summary, this is because the way the paper is presented suggests introduction of a novel method, which heavily borrows existing techniques and lacks further theoretical insights, and the current writing making it hard to envision a major change in scope (from a supposedly novel method to the new contribution of exhaustive empirical evaluation in favor of simpler baselines). More detail on the latter statement is provided below.

**Justification For Why Not Higher Score:**

Following the above argumentation, the specific rationale behind recommending the paper to be rejected in its current form is that the AC believes a major revision will significantly strengthen the paper and through improved presentation enhance its future impact for the community.

The reviews and rebuttal seem to have exposed that the paper will eventually contain a much stronger contribution than the adoption of a well-known gradient based continual learning inspired memory technique into federated learning. That is, as the authors themselves acknowledge, observing that continual federated learning techniques are getting ever more complicated, yet their experimental foundation seemingly being outperformed through simpler adopted baselines, is a worthy contribution of characterized exhaustively. Through the rebuttal it is visible that many steps are being taken to make the latter message the main contribution, with several ablation experiments, new comparisons, detailed investigations for choice of algorithm/memory buffer being actively developed. As such, an envisioned accepted paper might look more like prior works on e.g. GDUMB (which contained a similar memory related exhaustive empirical analysis for continual learning).

However, as much as the AC foresees the above to eventually come to fruition, such a change in paper structure and addition of several tables, analyses, and ablations, will also incur heavy changes to the current paper and its composition. These have not yet been included in the paper and are scattered across several rebuttal "boxes", where already addressed. For this reason, the AC recommends a major revision of the "story" the paper aims to tell, supported with the extensive experimentation that the authors have begun already. In summary, this includes the analysis wrt other recent methods, investigation of memory buffer considerations, analysis of communication costs, and the currently still unaddressed raised concerns on privacy leakage.

**Justification For Why Not Lower Score:**

N/A

---

### Decision · Program_Chairs · 2024-01-16

Reject